# Evaluation of the operational MODIS cloud mask product for detecting cirrus clouds

Żaneta  Nguyen Huu[1,2,3], Andrzej Z. Kotarba[4], and Agnieszka Wypych[1]

[1]Jagiellonian University, Institute of Geography and Spatial Management, Gronostajowa St 7, 30-387 Kraków, Poland
[2]Jagiellonian University, Doctoral School of Exact and Natural Sciences, Prof. St. Łojasiewicza St 11, PL30348, Cracow, Poland
[3]Institute of Meteorology and Water Management – NRI, Department of Satellite Remote Sensing, P. Borowego St 14, PL30215, Cracow, Poland
[4]Space Research Centre, Polish Academy of Sciences, Bartycka 18A, PL00716, Warsaw, Poland

Correspondence: Żaneta  Nguyen Huu (zaneta.nguyen_huu@uj.edu.pl)

**Abstract.** All clouds influence the Earth's radiative budget, with their net radiative forcing being negative. However, high-level clouds warrant special attention due to their atmospheric warming effects. A comprehensive characterization of cirrus clouds requires information on their coverage, which can be obtained from various data types. Active satellite sensors (lidars) are presently the most accurate source for cirrus data, but their usefulness in climatological studies is limited (the narrow view and 16-day repeat cycle yield only ~20 observations per year per region, often insufficient for climatological studies). On the contrary, passive data, which has been available for the past 40 years with sufficient temporal resolution for climatological research, are less effective at detecting cirrus clouds compared to active vertical profiling sensors. In this study, we assessed the utility of Moderate Resolution Imaging Spectroradiometer (MODIS) standard products for creating a cirrus mask by validating them against Cloud-Aerosol Lidar with Orthogonal Polarization (CALIOP) data. Our objective was to determine how well the operational cloud mask from the MODIS Science Team can be used to infer the presence of cirrus clouds relative to data products derived from the highly sensitive CALIOP instrument by the Cloud-Aerosol Lidar and Infrared Pathfinder Satellite Observations (CALIPSO) Science Team.

Using CALIOP data as the reference, we evaluated six tests for cirrus and high clouds detection considered in MODIS cloud masking algorithm and their combination (all tests consolidation, ATC). Additionally, we applied classifications based on the cirrus definition from the International Satellite Cloud Climatology Project (ISCCP), which rely on retrieved MODIS cloud-top properties. These were used not as detection tests, but as a classification scheme for comparative purposes. All other tests were applied directly to MODIS radiances.

Study revealed that the ATC test was the most effective resulting with the overall accuracy of 72.98% (probability of detection 80.9%, false alarm rate 34.9%, Cohen's $\kappa$ 0.46) during daytime and 59.50% at night (probability of detection 25.5%, false alarm rate 6.9%, Cohen's $\kappa$ 0.19). However, its effectiveness was notably reduced during nighttime compared to daytime. We conclude that the MODIS operational Cloud Mask after being modified into the ATC test is moderately suitable for creating a mask of high-level clouds, and only daytime. During the night-time MODIS ATC fails to reliably report the presence of cirrus.

# 1 Introduction

Clouds are indispensable to Earth's environmental systems and human life, influencing weather, climate, water distribution, ecosystems, and various human activities. They affect the Earth's radiation budget, with a net radiative forcing of approximately -20 Wm$^{-2}$ (Boucher et al., 2013), which results in an overall cooling effect on the planet. Nevertheless, special attention should be paid to high-level clouds - according to the WMO, high-level clouds include Cirrus, Cirrocumulus, and Cirrostratus (WMO, 1977) - commonly referred to as cirrus. Those clouds play a complex role in climate regulation. The relation between cirrus particles (size, shape and albedo) and Earth's radiation budget has been examined (Kinne and Liou, 1989; Macke et al., 1998; Mishchenko et al., 1996; Stephens et al., 1990; Zhang et al., 1994, 1999), resulting in a general conclusion that cirrus play an important role and can warm the atmosphere. They typically have a base above about 8,000 m and consist of small ice crystals. Due to their unique properties - such as altitude, temperature, effective particle size, surface thermal contrast, ice water path, and optical depth (Ackerman et al., 1988; Stephens et al., 1990; Stephens and Webster, 1981), they differ from low- and mid-level clouds in their effect on the Earth's radiation budget. Specifically, cirrus clouds allow shortwave radiation to reach the surface while reducing outgoing longwave radiation, thereby contributing to warming. Globally, it's been estimated that cirrus clouds have a net warming effect of 35.5 Wm$^{-2}$ (Campbell et al., 2016; Kärcher, 2018; Lolli et al., 2017; Oreopoulos et al., 2017) in part because they trap and reduce outgoing longwave radiation more efficiently than they reflect solar radiation back to space. Furthermore, cirrus clouds can alter the radiative forcing of other cloud types. For example, when medium and low clouds co-occur, their combined radiative effect is -18.8 Wm$^{-2}$, but the additional presence of cirrus raises this effect to 50.8 Wm$^{-2}$ (Oreopoulos et al., 2017).

A description of cirrus cloud properties is incomplete without information about their coverage. Most studies have focused on total cloud cover, but some have also examined high-level cloudiness. The global frequency of cirrus occurrence is estimated to range between 17% and 42%. Research conducted using high-resolution satellite data indicates that global cloud coverage is approximately 66% to 74%, with 40% of all clouds classified as high-level clouds (Sassen et al., 2008; Stubenrauch et al., 2010). Numerous studies have explored changes in high-level cloud coverage. However, those relying on satellite data often do not address cirrus clouds over sufficiently long periods—at least 30 years, as recommended by the WMO. Conducting such long-term studies and identifying suitable data sources remain significant challenges.

Given the critical role of cloud cover, especially cirrus, observing clouds is of considerable importance. Historically first method is visual observation from ground-based meteorological stations, which is simple and provides long time series data. However, this method has limitations, including difficulty in detecting high-level clouds due overlapping clouds at multiple altitudes, perspective distortions near the horizon, and the optical thinness of cirrus clouds. Studies have shown that under optimal conditions, the probability of visually detecting cirrus clouds ranges from 44% to 83% during the day and from 24% to 42% at night. When clouds at all levels are present, detection probabilities drop to 47%–71% during the day and 28%–43% at night (Kotarba and Nguyen Huu, 2022).

Modern cloud climatologies benefit from satellite remote sensing. Initially, this information was obtained from various imagers, sounders, and radiometers, which utilize passive cloud detection methods (involving detecting natural radiation emitted

or reflected by objects, such as clouds, without actively sending out signals). Researchers such as Ackerman et al. (2008); Amato et al. (2008); Chen et al. (2002); Frey et al. (2008, 2020); Gu et al. (2011); Kotarba (2016); Liu et al. (2004); Minnis et al. (2008); Murino et al. (2014); Musial et al. (2014); Tang et al. (2013) have contributed to these studies. An example of passive sensor can be MODIS, which is a key instrument aboard the Terra and Aqua satellites.

Active remote sensing technology, in contrast, relies on its own signal, directing it at an object and analysing the response. This allows active sensors, for instance CALIPSO's lidar, CALIOP, to operate day and night with similar efficiency in cloud detection. Active profiling instruments like CALIOP, which provide high-resolution vertical profiles of aerosols and clouds, have limitations, including a narrow field of view. This narrow view, combined with a long 16-day repeat cycle, results in only about 20 observations per year of the same region, which is challenging and sometimes insufficient for climatological studies (Kotarba and Nguyen Huu, 2022).

To standardize cloud classification and ensure consistency, the International Satellite Cloud Climatology Project (ISCCP) developed a system based on cloud height and optical thickness, providing a systematic framework for studying cloud types and their variability across regions and over time. This classification is crucial for advancing climate modelling, weather forecasting, and research on cloud-climate interactions. The ISCCP classification was applied to MODIS data, and its effectiveness in detecting cirrus clouds was also evaluated. In this study, we refer to the ISCCP classification not as a detection method, but as a widely accepted climatological framework based on retrieved parameters.

While active sensors like CALIOP remain the most reliable source of cirrus data (e.g., Heidinger and Pavolonis (2009)), their potential for building long-term climatologies is limited. In contrast, passive data have been available for over 40 years, offering temporal coverage suitable for climatological research. One example of such sensors, although collecting data for over 20 years rather than 40, is MODIS, whose capabilities for detecting cirrus clouds are limited compared to those of active vertical profiling sensors.

In this paper, we use cirrus characterizations from CALIOP data to explore the potential for creating a cirrus mask from the operational MODIS cloud data products. Our objective is to determine how well the MODIS products can be used to identify cirrus clouds compared to CALIPSO. Specifically, we aim to assess whether the existing MODIS cloud detection tests used in the generation of the MYD35 operational data can be re-purposed for cirrus cloud masking without the need to develop a new cirrus detection algorithm.

## 2 Data and methods

In this study, we use active sensor data for validating passive-based information for determining the presence of cirrus (for the sake of clarity, throughout this manuscript, all high-level clouds will be called as cirrus). The active sensor data was collected by the CALIOP lidar aboard the CALIPSO satellite, while the passive data was obtained from the MODIS multi-band radiometer on the Aqua satellite. The concept behind achieving the research objective was based on collocation of those two datasets in time and space. In both instances, cirrus clouds are the same physical phenomenon; however, the distinction arises from the

varying sensitivities of the detection instruments employed, with optical thickness serving as a crucial parameter. CALIPSO is capable of identifying cirrus clouds with an optical thickness as low as approximately 0.01, while MODIS generally detects them when the optical thickness is at least of 0.4 to 0.5 (e.g. Menzel et al. (2015)). Data for the year 2015 were analyzed on a global scale, comprising 136,272,209 collocated MODIS-CALIPSO observations. The primary requirement was to obtain a sufficiently large sample of CALIPSO-MODIS match-ups across different seasons and geographic regions, which necessitated one complete year of global observations. Therefore, 2015 was chosen arbitrarily.

## 2.1 MODIS data

MODIS, an advanced instrument aboard NASA's Terra and Aqua satellites, acquires data across 36 spectral bands, spanning wavelengths from visible to thermal infrared (0.4 to 14.4 $\mu$m). Its passive sensors rely primarily on naturally available energy: solar energy reflected from objects or absorbed and re-emitted (e.g. Ackerman et al. (1998)). MODIS provides data at various spatial resolutions - 250 m, 500 m, and 1 km - with a swath width of 2,330 km, enabling it to observe the entire Earth twice daily, one observation during the day and one at night. Cloud detection results are stored in the 48-bit "Cloud Mask" product, known as MYD35 for Aqua, while corresponding cloud properties can be found in MYD06 dataset. As an imager, MODIS provides column-integrated radiances, which limits its ability to retrieve cirrus-specific information. For this research, we used Collection 061 data, which is available in 5-minute granules at a spatial resolution of 1 km per pixel (at nadir). Each MYD35 and MYD06 file is paired with a MYD03 "Geolocation file" product that contains longitude and latitude information for each individual cloud mask IFOV (instantaneous field of view, Guenther et al. (2002).

### 2.1.1 The MODIS Cloud Mask product

The MODIS Cloud Mask product is a Level 2 dataset produced at spatial resolutions of 1 km and 250 m (at nadir). The cloud masking procedure was described in details by Ackerman et al. (1998); Frey et al. (2008); Baum et al. (2012). The algorithm utilizes a sequence of visible and infrared threshold and consistency tests to determine the confidence level that an unobstructed view of the Earth's surface is achieved.

The primary MODIS routine for identifying clouds is the MODIS Cloud Mask (product MOD35), which applies a series of spectral threshold tests to each pixel. The cloud mask algorithm does not explicitly label cloud type (no specific "cirrus" output flag); instead, it provides a confidence level that the pixel is cloudy or clear.

However, certain tests within the algorithm are specifically designed to detect high thin clouds like cirrus. It particularly applies to tests using the spectral band centred at 1.38$\mu$m, a MODIS-introduced wavelength for cirrus detection (Gao and Kaufman, 1995). In this research, we considered 6 individual cloud detection tests of MODIS cloud mask, which – according to the cloud mask detection algorithm (Ackerman et al., 1998) – have a potential for cirrus or high cloud detection:

– Thin Cirrus test (SOLAR) – the solar channels in MODIS cover a range of wavelengths primarily in the visible and near-infrared spectrum (0.4 to 2.5 $\mu$m). This test uses the solar range to set the confident clear and middle thresholds to

define the range of expected reflectances from thin cirrus. It indicates that a thin cirrus cloud is likely to be present. Test is only applied during daytime.

- Thin Cirrus test (IR) – the purpose of this test is detecting thin cirrus clouds. Channels used for this test are 11 $\mu$m an 12 $\mu$m (infrared (IR) range), incorporated to the split window technique. This test leverages the fact that cirrus clouds, composed of ice, are more transparent at 11 $\mu$m than at 12 $\mu$m, resulting in a positive BTD (Brightness Temperatures Difference) signature.

- High Cloud Test (BT13.9) – applying $CO_2$ absorption channels (around 14 $\mu$m) is a simple technique got from the $CO_2$ slicing method (suitable for determining middle and upper troposphere ice clouds heights and effective amounts). This test is useful for high-level cloud detection, while it can reveal clouds above 500 hPa – it helps identify high clouds by detecting colder cloud tops using the $CO_2$ absorption band.

- High Cloud Test (BT6.7) – test designed for detecting thick high clouds. Starting from the ground level, the 6.7 $\mu$m radiation emitted by the surface or low clouds is absorbed in the atmosphere, therefore the signal is not received by an instrument. The water vapor in layer in the atmosphere between 200 hPa and 500 hPa is the only source of the 6.7 $\mu$m radiation in clear-sky observation. Thick clouds placed above or near the 200 hPa level can be distinguish from clear sky or lower clouds.

- High Cloud Test (BT1.38) – the 1.38 $\mu$m channel lies in the strong water vapor absorption region. That results in obscuration of the most of Earth's surfaces, as well as attenuation of reflectance from low- and mid-level clouds. Pixels with this test applied, reveals high-level thin clouds as brighter. Unfortunately, the test has certain limitations, including its applicability to nighttime conditions, polar regions, midlatitude winters, and high elevations..

- High Cloud Test (BT3.9-12.0) – the 3.9-12.0 $\mu$m BTD test is specifically designed for nighttime observations over land and polar snow/ice surfaces. It is effective in distinguishing between thin cirrus clouds and cloud-free conditions and exhibits relative insensitivity to the atmospheric water vapor content (Hutchinson and Hardy, 1995).

Additionally, we independently developed a unified approach to combine all tests, which we termed **All Tests Consolidation** (ATC). If any ($\exists$ - there is at least one) of the nine tests (t) detected cirrus clouds, the output flag (OF) was set to indicate the presence of cirrus. Conversely, if no cirrus clouds were detected by any of the tests ($\forall$ - for every), provided they were all conducted, no cirrus flag was set. In cases where all nine tests returned a value of 9, indicating missing or unavailable data, the output flag was also set to 9, to explicitly represent the absence of valid input across all tests. This allows the ATC approach to distinguish between a confirmed absence of cirrus and a lack of information.

$$\text{ATC}_{\text{OF}} = \begin{cases} 1, & \text{if } \exists i \in \{1,\ldots,9\},\ t_i = 1 \\ 0, & \text{if } (\forall i,\ t_i \in \{0,9\}) \wedge (\exists j,\ t_j = 0) \\ 9, & \text{if } \forall i,\ t_i = 9 \end{cases} \tag{1}$$

ATC is essentially an adaptation of the MYD35 approach, but it is limited to tests that provide insights specifically about cirrus clouds.

### 2.1.2 The MODIS Cloud Product

As described by Menzel et al. (2015) the MODIS Cloud Product uses a combination of infrared and visible techniques to determine cloud physical and radiative properties. It derives cloud-particle phase, effective particle radius, and optical thickness from visible and near-infrared radiances, and indicates cloud shadows. Infrared methods provide cloud-top temperature, height, effective emissivity, phase, and cloud fraction, both day and night, at 1-km-pixel resolution. For Aqua satellite, dataset is called MYD06.

In addition to the ready-to-use MODIS tests (Section 2.2.1), other criteria can be applied using data available from MODIS and CALIOP. For instance, the ISCCP's definition of cloud types. By examining visible and infrared radiances from geostationary and polar-orbiting meteorological satellites and making assumptions about cloud layering, thermodynamic phases, and properties, ISCCP characterizes a cloudy satellite pixel using the column visible optical depth (COT) and the cloud-top pressure (CTP) of the highest cloud layer. This information is used to classify different cloud types as shown in the figure 1 (Rossow and Schiffer, 1991).

COT and CTP is also available for MODIS, within MYD06 standard product, and we used it to generate cirrus masks according to ISCCP definition. We considered two variants of the mask, defining cirrus as:

- a cloud with an optical thickness less than 3.6 and a top pressure below 440 hPa (hereinafter ISCCP3.6 test),

- a cloud with an optical thickness less than 23 and a top pressure below 440 hPa (hereinafter ISCCP23 test).

It is important to clarify that the ISCCP-based classes used here are not interpreted as detection algorithms in the same sense as the MODIS cloud mask tests. Instead, they serve as a classification of cloud type based on retrieved cloud optical thickness and top pressure, following established ISCCP thresholds. These classifications are included for reference as they are widely used in long-term cloud climatology studies.

### 2.2 CALIOP data

CALIOP provides atmospheric profiles with vertical resolutions ranging from 30 m below 8.2 km to 180 m above 20.1 km, and 60 m between these altitudes (Winker et al., 2006). This capability allows for clear distinction between cirrus and lower

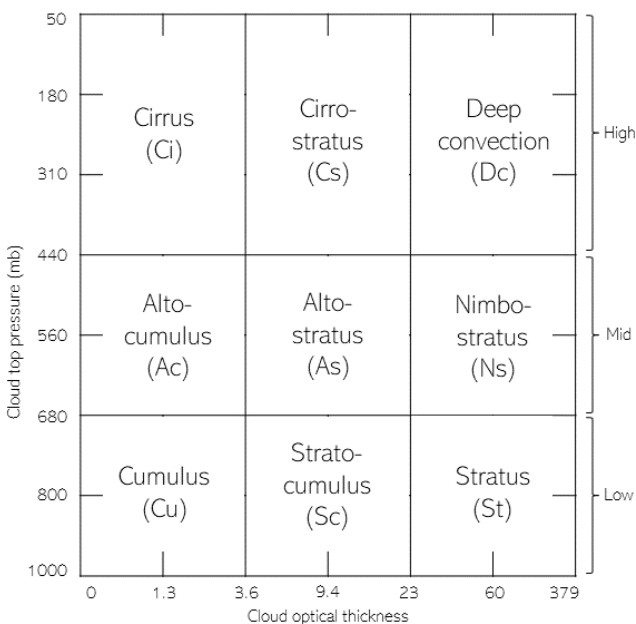

**Figure 1.** The distribution of cirrus clouds according to the evaluation.

cloud layers, making CALIOP excellent for cirrus detection. Furthermore, lidar can detect cirrus clouds with an optical depth as low as 0.01 (Vaughan et al., 2009), a capability beyond the reach of passive imagers (Ackerman et al., 2008). Being an active sensor, lidar offers similar effectiveness in cloud detection both daytime and nighttime, or even higher during night, when backscattered light does not interfere with diffused solar radiation (McGill et al., 2007)

In this research, the lidar level-2 cloud layer at 5-km horizontal resolution, version 4.20 (CAL_LID_L2_05kmCLay-Standard-V4–20) product was used. As described by Liu et al. (2009) and Vaughan et al. (2009) this product reports cloud layers and cloud type information, with cirrus as a separate class (categorized as type 6). In the CALIPSO system, cirrus clouds are detected using the SIBYL (Selective Iterated Boundary Location) and CAD (Cloud-Aerosol Discrimination) algorithms. The

SIBYL algorithm identifies cloud layers based on enhanced backscatter signals in CALIOP lidar profiles. Subsequently, using a probabilistic approach, the CAD algorithm differentiates between clouds (including cirrus) and aerosols. The depolarization ratios for cirrus clouds are higher than those for water-based clouds, enabling their identification. Additionally, CALIOP provides an information about the cloud base and top altitudes, allowing for determination of their position in the atmosphere. In CALIOP data, cirrus clouds are identified as high-altitude (cloud-top pressure < 440 hPa) and transparent layers - meaning

the lidar can detect the surface or a lower atmospheric layer beneath. If a layer is opaque, even at high altitude, it is not classified

as cirrus. This ensures that only optically thin, high-level ice clouds are labeled as cirrus. Clouds above the tropopause, such as polar stratospheric clouds, were excluded as they constitute a separate feature type in CALIPSO data. The quality of CALIOP's detection is reflected in CAD score, which ranges from -100 to 100. Value -100 indicates high confidence of aerosol detection, while a value of 100 indicates high confidence in cloud detection. A medium value (0) signifies equal probability that the feature is a cloud or aerosol. To mitigate misclassification, particularly over dust regions (e.g., the Sahara), the CAD algorithm dynamically adjusts depolarization thresholds (Liu et al., 2009; Vaughan et al., 2009). In this study, we used only observations with a CAD score higher than 80. The optical depth is also provided in this (CAL_LID_L2_05kmCLay-Standard-V4–20) CALIOP product.

For the purpose of this research, we consider CALIPSO as the reference for cirrus clouds detection.

## 2.3 Matching datasets

In order to achieve the goal of this study, MODIS and CALIOP data were collocated in space and time. It was possible because Aqua and CALIPSO operated for 12 years (2006-2018) as a part of satellite constellation commonly known as the Afternoon Constellation. Members of the constellation used sun-synchronous polar orbits with 16-day revisit cycle, and with equatorial crossing time at 13:30 local solar time (ascending node). CALIPSO followed the Aqua spacecraft by approximately one minute (e.g. Stephens et al. (2018)), enabling quasi-simultaneous observation of the same part of the atmosphere, as 1 km ground track of CALIOP always overlapped with 2,330 km wide imagery of MODIS.

Collocating MODIS with CALIOP has been frequently used to validate reliability of MODIS datasets, or to developed a new, joint imager-lidar atmospheric products (Baum et al., 2012; Holz et al., 2009; Kotarba, 2020; Sun-Mack et al., 2014; Wang et al., 2016; Xie et al., 2010). Either 333 m, 1 km, or 5 km lidar data may be considered, however only 1 km and 5 km products offers cloud type classification. Additionally, only the 5 km product informs about cloud optical thickness per cloud layer, and provides superior cirrus detection due to higher sensitivity (noise level decreases as more profiles is integrated into retrieval). From the geometry point of view, a 5 km profile is an aggregation of five consecutive 1 km profiles, and the geo-coordinates of the central one are saved as representative for 5 km profile. It possess a challenge when MODIS and CALIOP are to be matched: one 5 km profile of CALIOP only can be accurately matched to one 1 km MODIS pixel, while 5 km data actually covers five MODIS pixels. To overcome this problem we matched CALIOP with MODIS using non-aggregated, 1 km data, and only then assigned 5 km data to already collocated MODIS-CALIOP pairs. As a result, one 5 km profile of CALIOP was used to characterize five MODIS pixels.

Aqua and CALIPSO ground tracks are offset by 100-120 km at the equator (decreasing towards the poles). It means, that they observe the atmosphere from slightly different angles, causing a parallax shift. We did not correct the data for parallax, as its impact only would be observed close to the edges of clouds, which are small fraction of all observations, or for investigating dynamically-changing cloud top properties (Wang et al., 2011) which was not the case of our investigation.

This study relied on MODIS-CALIOP observations for 2015, and the year was selected arbitrary, as the only requirements was to consider a relatively large (year-long) sample of global observations of clouds. Eventually, our database consisted of 136,272,209 paired MODIS-CALIOP observations; the average spatial distance between geometrical centers of matched

MODIS pixel and CALIOP profile was 444 m (std. dev. = 231 m), while the average temporal separation reached 84 seconds (std. dev. = 12 seconds).

The final, aggregated MODIS–CALIOP statistics were compiled into global maps, each with a spatial resolution of 5° in both longitude and latitude.

## 2.4 Evaluation of MODIS data

We consider a test as a useful for cirrus detection whenever it results with a high agreement with the reference data (CALIOP). The degree of agreement was calculated using a confusion matrix approach for a binary classifier ('cirrus' and 'no cirrus' classes). The approach compares the model's predictions (MODIS performance) against the actual results (CALIPSO detection of cirrus). Table 1 provides list of statistical indices we used as a measure for MODIS-CALIPSO agreement in cirrus detection.

**Table 1.** Confusion matrix-based measures used for assessing MODIS agreement with CALIPSO in this study

| Abbreviation | Full name |
| --- | --- |
| TP | True Positives |
| FP | False Positives |
| TN | True Negatives |
| FN | False Negatives |
| ROP | Rate of Observations Performed |
| POD | Probability of Detection |
| FAR | False Alarm Rate |
| OA | Overall Accuracy |
| $\kappa$ | Cohen's kappa $\kappa$ coefficient |
| PE | Expected agreement |
| $n$ | Number of elements in the set |

The structure of confusion matrix is presented in Table 2. and includes the following elements:

– True Positives (TP): The count of cases where MODIS accurately identified the existing (according to CALIOP) cirrus.

– False Positives (FP): The count of cases where MODIS incorrectly identified the high-level cloud, meaning it detected cirrus presence when it was actually absent.

– True Negatives (TN): The count of cases where MODIS correctly did not detect the presence of the cloud.

– False Negatives (FN): The count of cases where MODIS overlooked the cirrus occurrence.

Every result undergoes thorough validation through different parameters estimation using feature-based statistics (Stanski et al., 1989). To describe the data accuracy, probability of detection (POD) characteristics [1] and false alarm rate (FAR) [2]

**Table 2.** Confusion matrix

| CALIPSO (reference data) | Cirrus | No Cirrus |
|---|---|---|
| MODIS Cirrus | True positive (TP) | False positive (FP) |
| MODIS No Cirrus | False negative (FN) | True negative (TN) |

metrics were calculated:

Probability of detection (POD) – is a metric used to assess the effectiveness of a detection system. In the context of cloud detection, POD indicates how well the detection algorithm correctly identifies the presence of clouds when they are actually present. A higher POD value signifies better performance of the detection system.

$$POD = TP/(TP + FN) \tag{2}$$

False alarm rate (FAR) – is a metric that measures the frequency of incorrect positive detections by a system. In the context of cloud detection, a lower FAR indicates a more accurate system, with fewer instances of falsely identifying clouds when they are not present.

$$FAR = FP/(FP + TN) \tag{3}$$

The incident frequencies within the matrix enabled the identification of two more diagnostic measures: Overall accuracy (OA) – is a metric that measures the proportion of correct predictions made by a detection system out of all predictions. In cloud detection, higher overall accuracy indicates that the system effectively identifies both the presence and absence of clouds correctly.

$$OA = (TP + TN)/n \tag{4}$$

Cohen's $\kappa$ – Cohen's kappa is a statistical metric used to assess the degree of agreement between two raters or classification methods (Cohen, 1960). Its scale ranges from -1 to 1, where a value of 1 represents perfect agreement, 0 indicates agreement no better than chance, and negative values indicate agreement worse than chance. In cloud detection, a higher kappa value indicates stronger agreement between the detected presence of clouds and their actual presence, while considering the possibility of random agreement.

$$\kappa = (OA - PE)/(1 - PE) \tag{5}$$

where

PE – expected agreement

$$PE = [(TP + FP)(TP + FN) + (TN + FP)(TN + FN)]/n^2 \qquad (6)$$

$$n = TP + FP + FN + TN \qquad (7)$$

The accuracy of high-level cloud detection was evaluated using the aforementioned metrics, differentiated by day and night,
latitude, cloud optical depth, the number of detected cloud layers, and land classification. This assessment was conducted for
the entire year 2015, as well as specifically for January and July (those two months are presented to exemplify the characteristics
of two distinct seasons).

## 2.5 Bootstrap sampling

Due to the nature of cirrus cloud occurrences (18.7% in 2015, see Section 3), we can assume that the data sample will be im-
280 balanced and one class (without cirrus) significantly outnumbers the other. Therefore, for such data, the appropriate statistical
method to apply is bootstrap sampling (Efron, 1980).
The balancing of the sample stems from the issue of class imbalance, potentially skewing the statistical analysis and leading to
biased results. To mitigate this, the bootstrap method is employed to artificially balance the dataset. This involves resampling
the data with replacement, to ensure that each class has a comparable number of instances. By doing so, the analysis can
yield more accurate results, rather than being dominated by the majority class. When a sample is drawn from a population,
the statistical measures derived from that exhibit sampling variability. The fundamental concept of bootstrap revolves around
resampling the original dataset with replacement to generate multiple bootstrap samples. In our study, for 1000 iterations,
we selected a sample with replacement that included all observations indicating the presence of cirrus clouds (according to
CALIPSO), as well as an equal number randomly drawn from the remaining observations. Each time, the previously described
measures were calculated. After performing these calculations 1000 times, the average of these measures was computed.
To demonstrate the concept of bootstrap sampling, we conducted a simple experiment using a dataset consisting of 100 obser-
vations. Of these, 15 correspond to cirrus clouds (positive class), and 85 correspond to non-cirrus clouds (negative class). Given
the significant class imbalance, many models tend to favor the majority class, leading to overly optimistic accuracy metrics.
For example, a naive model that predicts "non-cirrus" for all observations achieves an overall accuracy (OA) of 85%, correctly
classifying all negative instances while entirely disregarding the minority class:

$$OA = (TP + TN)/n = (0 + 85)/100 = 0.85 (85\%) \qquad (8)$$

To mitigate this imbalance, we applied bootstrap sampling to generate a balanced dataset through resampling with replacement,
ensuring an equal number of positive and negative instances (e.g., 15 cirrus and 15 non-cirrus cases). When the same naive

model was applied to the balanced dataset, the overall accuracy dropped to 50%, highlighting the model's inability to correctly classify the minority class:

$$OA = (TP + TN)/n = (0 + 15)/30 = 0.50 (50\%) \tag{9}$$

This experiment illustrates how bootstrap sampling can reveal the shortcomings of models trained on imbalanced datasets, offering a more accurate and realistic assessment of model performance.

The bootstrap has been already widely used among climatological studies. It has been employed to, among others, estimate confidence interval (Jolliffe, 2007), forecast storm track (Wilks et al., 2009), project future climate (Orlowsky et al., 2010), verify potential predictability of seasonal mean temperature and precipitation (Feng et al., 2011), study seasonal prediction of drought (Behrangi et al., 2015), inspect macrophysical properties of tropical cirrus clouds (Thorsen et al., 2013), evaluate sampling error in TRMM/PR rainfall products (Iida et al., 2010).

## 3 Cirrus distribution in 2015 according to CALIOP

Before conducting an analysis to assess the agreement in high-level cloud detection between CALIOP and MODIS data, we examined the cirrus coverage in 2015 according to reference data (CALIOP). The cirrus cloud mask was generated by applying a condition that classified each 5-degree latitude-longitude grid cell based on the proportion of observations identified as cirrus. Specifically, the number of cirrus observations and non-cirrus observations within each 5-degree grid cell were counted. The percentage of cirrus observations for a given 5-degree grid cell was a fraction of observations with cirrus detected to all observations.

This approach ensures that the mask reflects the relative frequency of cirrus clouds within each 5-degree grid cell, providing a spatially resolved representation of their distribution.

The distribution of cirrus clouds (Fig. 2.) varies globally and is affected by factors such as latitude and atmospheric dynamics. According to Sassen et al. (2008), the total frequency of cirrus clouds from 15 June 2006 to 15 June 2007 was reported as 16.7%, compared to 18.7% observed in our study for 2015. However, according to the research by Kotarba and Nguyen Huu (2022), annual mean values of cloud amount, derived from CALIPSO, can vary significantly (over 10 p.p. (percentage points)) between years due to sampling frequency.

Cirrus clouds are more frequently observed at night, particularly in tropical and mid-latitude regions, with their occurrence peaking around midnight and reducing during the day also according to Noel et al. (2018). Moreover, frequencies of stratospheric cirrus clouds measured by CALIPSO from 2006 to 2012 detected at nighttime are 2-3 times higher those detected during daytime (Zou et al., 2020). Nevertheless, the day-night difference observed in Sassen et al. (2008) study was smaller than in ours, with values of 15.2% during the day and 18.3% at night, compared to 13.2% (Fig. 2a.) and 23.3% (Fig. 2b.), respectively, in our analysis. These differences may stem from the use of different (earlier) versions of source datasets and the application of different criteria for selecting and screening cloud layer data. The higher detectability of nighttime cirrus clouds may also be attributed to reduced noise in lidar signals under nighttime conditions (the lack of solar background). Additionally,

the differences might also reflect more intense convective activity and increased formation of cirrus clouds during the night. However, diurnal differences in cirrus occurrence are complex, the artificial diurnal difference, driven by the varying levels of background noise during the day and night, likely outweighs the actual diurnal variations.

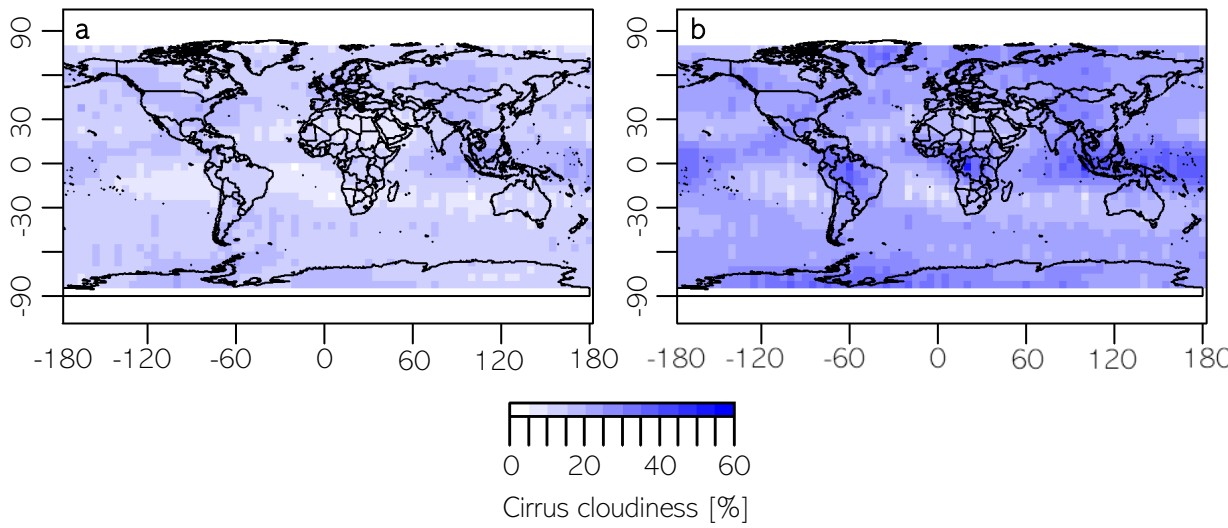

**Figure 2.** The distribution of cirrus clouds according to the evaluation - daytime (a) and nighttime (b)

In our study, near the equator, especially within the tropical belt, cirrus cloud cover exhibits peak values throughout the year, reaching approximately 35% during nighttime and 20% during daytime. In certain locations, particularly during nighttime, the high-level cloudiness has been observed to exceed 50%. In the mid-latitudes of both hemispheres, the distribution of clouds varies, generally showing lower coverage compared to low latitudes, with approximately 10% during daytime and 20% at night. In polar regions, particularly above approximately 60° latitude, cirrus cloud cover tends to be higher than in mid-latitudes, with
nighttime coverage generally higher than daytime (Fig. 3.).

     Additionally, CALIOP measures the cloud optical thickness for individual layers as well as for the entire atmospheric column (Fig. 4.). When CALIOP detects multiple cirrus cloud layers, the COT values for all layers flagged as cirrus are summed. The mean cirrus COT was observed to be 0.72 during daytime and 0.84 at nighttime, indicating a notable increase in optical thickness at night. This can raise important question about the underlying cause of this difference. One possible explanation is
that the increased nighttime COT enhances the likelihood of cirrus cloud detection, as lidar systems like CALIPSO have greater sensitivity to optically thicker clouds, and even more so at night due to the absence of solar background and a higher signal-to-noise ratio. Consequently, this could lead to a higher observed cloud cover at night simply due to improved detectability rather than actual physical differences in cloud properties. However, this argument may not fully align with the observed data. Given

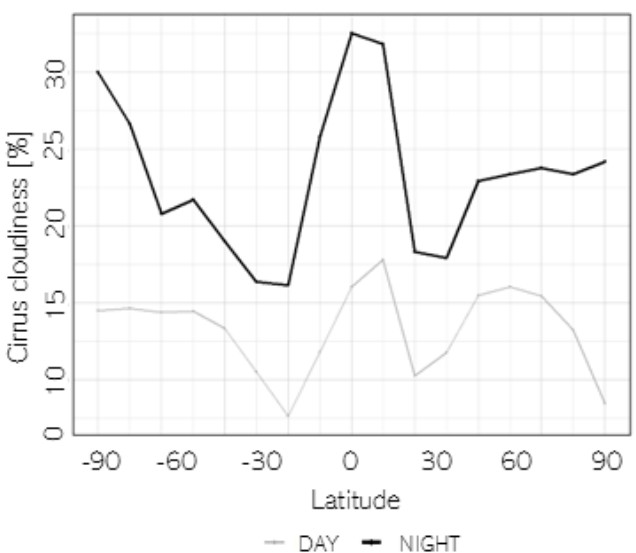

**Figure 3.** Cirrus coverage as a function of latitude (CALOP data)

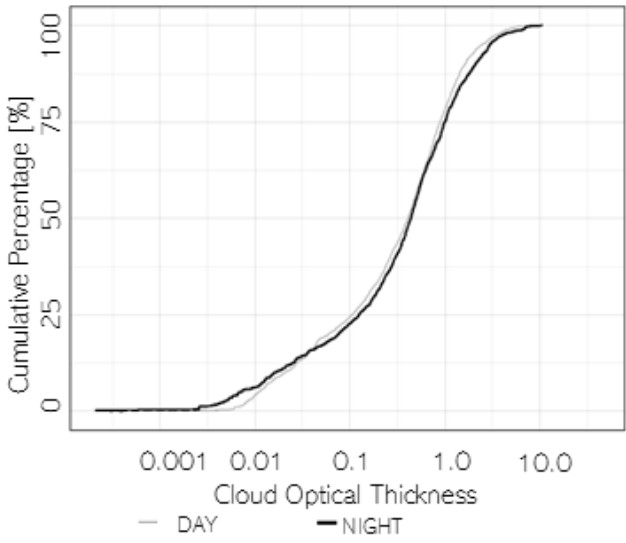

**Figure 4.** Cumulative ratio of cirrus clouds with respect to COT (CALIOP data)

that CALIOP is more sensitive at night, it could be expected to detect more thin clouds, potentially lowering the average COT

compared to daytime. The observed increase in nighttime cirrus COT could stem from multiple sources, including genuine

diurnal variability, retrieval algorithm behavior, or screening-induced bias. Further investigation would be needed to isolate their individual contributions. Alternatively, data filtering processes might contribute to the observed disparity.

## 4   Evaluation of MODIS data

Using CALIPSO data as the reference, nine methods for detecting cirrus clouds with MODIS data were evaluated. All tests
were applicable during daytime, whereas only five could be utilized at nighttime due to the requirement of solar illumination. The measures described in section 2 are presented in Table 3. The parameters that, in our opinion, precluded the use of the test are highlighted in bold. Additionally, they are preceded by the rate of observations performed (ROP) parameter, which is fraction of all observations under test.

During daytime, the first four methods (SOLAR, IR, BT13.9, BT6.7) exhibited notably low detection effectiveness (with POD
ranging between 0.33 and 15.79%), as well as low $\kappa$ coefficients (0.01-0.48). Although the test was performed on a relatively high proportion of observations (78.37% - 97.59%), with a low number of false alarms (FAR between 1.23% and 13.16%) and good overall accuracy (OA ranging between 48.61% and 53.80%), the poor detection capabilities (indicated by POD) rendered these data inadequate as reliable sources of information on the occurrence of Ci clouds. The differing parameters excluded tests BT3.9-12.0 and those with ISCCP criteria from consideration. The limited number of observations with available results
from these tests rendered them impractical for use.

 The two tests most effective globally were BT1.38 and ATC. With very similar parameters (POD, FAR, OA and $\kappa$) the ATC test demonstrated superiority due to a significantly higher number of available observations (78.37% vs 98.67%, respectively). Among the night tests, IR, BT13.9, and BT6.7 exhibited low detection capabilities (POD 0.60% - 10.59%), whereas the BT3.9-12.0 test was performed only on 38.09% of observations. As with the daytime tests, the ATC test proved to be the most suitable
for detection.

Considering that global statistics for January and July were not markedly different from the yearly averages (Tab. 3.), subsequent analyses were conducted using data from the entire year.

All statistical measures also were calculated for different latitudes (Fig. 5.).

The observed latitudinal variability can be attributed to the physical properties of the different radiation wavelengths used by
each channel, as well as their specific functions. Additionally, this variability is influenced by factors such as the spatial distribution of cirrus clouds and the varying illumination conditions across latitudes. For almost all of the tests we observe the ROP (Fig. 5a. & Fig. 5b.) decrease with the latitude increase. This is related to presence of solar illumination. The exception is ROP according to BT3.9-12.0 (which increase from 0% in tropics to almost 30% in polar region) and was specifically designed for nighttime observations over land and polar snow/ice surfaces. ROP for both tests using ISCCP criteria is equal.
The latitudinal distribution of POD during the day (Fig. 5c.) shows that ISCCP criteria most accurately detected cirrus clouds in the tropical regions (up to 75% for ISCCP23 and almost 100% for ISCCP3.6), with POD reduction with latitude decrease (to about 10% and 40% respectively). A similar pattern was observed i.e. for BT13.9 method, but with cirrus detection capabilities about 3 times inferior. Depending on the test, latitudinal variability of POD could be also higher for mid-latitudes (ATC), low

**Table 3.** Goodness-of-fit of cloud detection between MODIS and CALIOP. Bold - parameters that precluded the use of the test

| | Daytime | | | | | Nighttime | | | | |
|---|---|---|---|---|---|---|---|---|---|---|
| **Test** | **ROP [%]** | **POD** | **FAR** | **OA** | $\kappa$ | **ROP [%]** | **POD** | **FAR** | **OA** | $\kappa$ |
| SOLAR | 78.37 | **15.79** | 13.16 | 51.66 | **0.03** | **0.00** | NA | NA | NA | NA |
| IR | 83.32 | **12.56** | 4.37 | 53.80 | 0.48 | 73.98 | **10.59** | 3.27 | 54.94 | 0.52 |
| BT13.9 | 65.52 | **1.35** | 3.59 | 48.61 | -0.02 | 71.02 | **2.13** | 3.42 | 50.67 | **-0.01** |
| BT6.7 | 97.59 | **0.33** | 1.23 | 49.92 | **-0.01** | 91.44 | **0.60** | 1.58 | 50.23 | **-0.01** |
| BT1.38 | 78.37 | 77.76 | 28.28 | 74.71 | 0.49 | **0.00** | NA | NA | NA | NA |
| BT3.9-12.0 | **7.39** | 64.48 | 15.36 | 72.41 | 0.46 | **38.09** | 39.09 | 5.46 | 65.26 | 0.33 |
| ATC | 98.67 | 80.87 | 34.86 | 72.98 | 0.46 | 94.84 | 25.46 | 6.90 | 59.50 | 0.19 |
| ISCCP23 | **37.97** | 84.16 | 72.00 | 61.26 | **0.13** | **0.00** | NA | NA | NA | NA |
| ISCCP3.6 | **37.97** | 33.30 | 16.54 | 58.96 | **0.17** | **0.00** | NA | NA | NA | NA |
| | **January** | | | | | | | | | |
| | Daytime | | | | | Nighttime | | | | |
| **Test** | **ROP [%]** | **POD** | **FAR** | **OA** | $\kappa$ | **ROP [%]** | **POD** | **FAR** | **OA** | $\kappa$ |
| SOLAR | 74.84 | **15.08** | 13.50 | 49.28 | **0.02** | **0.00** | NA | NA | NA | NA |
| IR | 78.95 | **12.47** | 4.54 | 51.81 | 0.46 | 72.30 | **10.53** | 3.46 | 54.07 | 0.51 |
| BT13.9 | 67.59 | **1.66** | 3.66 | 46.28 | -0.02 | 72.26 | **2.36** | 3.32 | 49.65 | **-0.01** |
| BT6.7 | 97.95 | **0.23** | 1.09 | 49.68 | **-0.01** | 99.97 | **0.59** | 1.43 | 49.59 | **-0.01** |
| BT1.38 | 74.84 | 79.65 | 31.69 | 74.22 | 0.48 | **0.00** | NA | NA | NA | NA |
| BT3.9-12.0 | **7.02** | 56.89 | 13.50 | 69.48 | 0.41 | **41.19** | 35.00 | 3.80 | 64.37 | 0.30 |
| ATC | 98.98 | 80.23 | 34.17 | 73.03 | 0.46 | 99.98 | 23.38 | 6.12 | 58.63 | 99.98 |
| ISCCP23 | **38.55** | 84.27 | 68.88 | 64.10 | **0.17** | **0.00** | NA | NA | NA | NA |
| ISCCP3.6 | **38.55** | 33.38 | 14.58 | 59.27 | **0.19** | **0.00** | NA | NA | NA | NA |
| | **June** | | | | | | | | | |
| | Daytime | | | | | Nighttime | | | | |
| **Test** | **ROP [%]** | **POD** | **FAR** | **OA** | $\kappa$ | **ROP [%]** | **POD** | **FAR** | **OA** | $\kappa$ |
| SOLAR | 84.32 | **16.57** | 11.58 | 53.99 | **0.05** | **0.00** | NA | NA | NA | NA |
| IR | 92.26 | **11.99** | 3.76 | 54.17 | 0.49 | 68.77 | **10.02** | 2.61 | 57.81 | 0.56 |
| BT13.9 | 65.65 | **1.89** | 3.72 | 49.61 | -0.02 | 67.48 | **2.62** | 3.93 | 53.88 | **-0.01** |
| BT6.7 | 99.69 | **0.15** | 1.06 | 49.63 | **-0.01** | 81.30 | **0.84** | 1.96 | 52.06 | **-0.01** |
| BT1.38 | 84.32 | 74.97 | 22.06 | 76.52 | 0.53 | **0.00** | NA | NA | NA | NA |
| BT3.9-12.0 | **7.67** | 72.20 | 21.54 | 74.30 | 0.47 | **37.58** | 47.02 | 7.95 | 67.82 | 0.38 |
| ATC | 99.96 | 83.14 | 31.76 | 75.69 | 0.51 | 88.61 | 30.47 | 7.99 | 62.05 | 0.23 |
| ISCCP23 | **36.57** | 85.54 | 74.77 | 61.16 | **0.12** | **0.00** | NA | NA | NA | NA |
| ISCCP3.6 | **36.57** | 32.84 | 16.26 | 58.67 | **0.17** | **0.00** | NA | NA | NA | NA |

latitudes (test utilizing the solar radiation range), or remained relatively unchanged. There is no clear trend of increasing/decreasing POD with latitude during the night (Fig. 5d.; slightly more cirrus correctly detected for polar regions by IR, BT13.9 and BT3.9-12.0 tests). The mid-latitudes exhibit POD drop for BT6.7 test, and consequently ATC test.

Figure 5 (Fig. 5e. & Fig. 5f.) shows also the latitudinal variability of FAR. In the tropical regions most of the tests show peak of falsely reported cirrus clouds during daytime in equatorial region (with maximum exceeding 90% for ISCCP23 and 50% for ISCCP3.6). Additionally, BT1.38 test falsely detects cirrus more often with increasing latitude, which results in 'bimodal' FAR distribution with peaks in tropics (about 35%) and midlatitudes (75% for northern hemisphere and 30% for southern). A distribution resembling BT1.38 exhibited test ATC, but with an upward shift of about 10 percentage points. The ATC test exhibited a latitudinal distribution of false alarm rate (FAR) that closely resembled the pattern observed for the BT1.38 test,

but with values shifted upward by approximately 10 percentage points.

No significant differences were found between the equatorial and polar regions for all the tests for OA. For daytime the latitu-
dinal variation was more readily observable and varied (Fig. 5g. & 5i. vs Fig. 5h. & 5j.).

Considering the very high proportion of correctly detected cirrus clouds, the high overall accuracy and $\kappa$ coefficient (degree
of agreement between two classification methods), ATC test showed the highest agreement with CALIOP data. Additionally,
it covers nearly all observations in the test (96.7%) and shows relatively low variability of statistical measures across different
latitudes. Therefore, it can be used as a basis for studies evaluating cirrus cloud coverage in long term perspective.

To ensure the ATC test performs optimally under various conditions and to provide a comprehensive analysis, fit measures
were additionally evaluated for "number of layers found" (NLF, Fig. 6.) and IGBP (The International Geosphere–Biosphere
Programme, Table 4).

CALIOP data products allow to report up to 10 cloud layers within a profile. When multiple cloud layers overlap, the lidar
signal may be attenuated, potentially leading to underestimation of cloud detection. Our research evaluated the collocation of
MODIS data to the reference CALIOP data, segmented by the number of detected cloud layers excluding cirrus clouds. A zero
indicated that no other cloud layers were detected besides possible cirrus in a given profile. Both day and night observations
revealed a maximum of four additional cloud layers. Based on the test conducted, ROP either decreased (i.e. BT13.9 70%
to 30% at daytime or BT3.9-12.0 at nighttime), increased (7% to 25% at daytime for BT3.9-12.0), or remained stable with
an increasing number of cloud layers (Fig. 6a. & Fig. 6b.). For ATC test, no discernible trend was identified. No clear trend
could be observed for POD, both day and night (Fig. 6c. & Fig. 6d.). However, the distribution of the FAR parameter exhibited
a different pattern. In several tests, particularly the ATC test, the FAR value (Fig. 6e & Fig. 6f) significantly increased with
the number of detected cloud layers (from 9% to 78% during the day and from 1% to 15% at night for the ATC test). This
pattern suggests that for clouds with significant vertical development (i.e., those containing multiple layers), MODIS tended
to identify only the uppermost layer, mistakenly classifying it as the entire cloud profile. As a result, the increasing number
of falsely detected cirrus clouds, particularly in cases of non-cirrus layers (NLF), is reflected in the distributions of OA and $\kappa$.
Specifically, as the number of non-cirrus layers increases, both OA and $\kappa$ values decrease, for both day and night observations
(Fig. 6g., Fig. 6h., Fig. 6i. & Fig. 6j.).

The International Geosphere–Biosphere Programme defines ecosystem surface classifications. For purpose of this study, 17
IGBP groups were aggregated to 3 classes: water, land and snow (goodness-of-fit with respect to land classification is presented
in Table 4.). Bright surfaces like snow, ice deserts, or complex terrain with varying surface types can make it challenging to
distinguish clouds from the ground. The first noticeable aspect is the significantly lower ROP for snow compared to other
classes. Generally, the fit measures are similar to those in previous analyses. During the day, ATC test performs better over
water, whereas SOLAR test performs better over land. On the contrary, during nighttime, BT3.9-12.0 test performs better over
water, whereas ATC test performs better over land. The analysis with respect to NLF and land cover types confirmed that ATC
test is best suited for achieving the objective of this study. Therefore, the spatial distribution of the individual fit measures for
this test was examined (Fig. 7).

The spatial distribution reveals a very high ROP for both: daytime (Fig. 7a.) and nighttime (Fig. 7b.) for the entire Earth. The

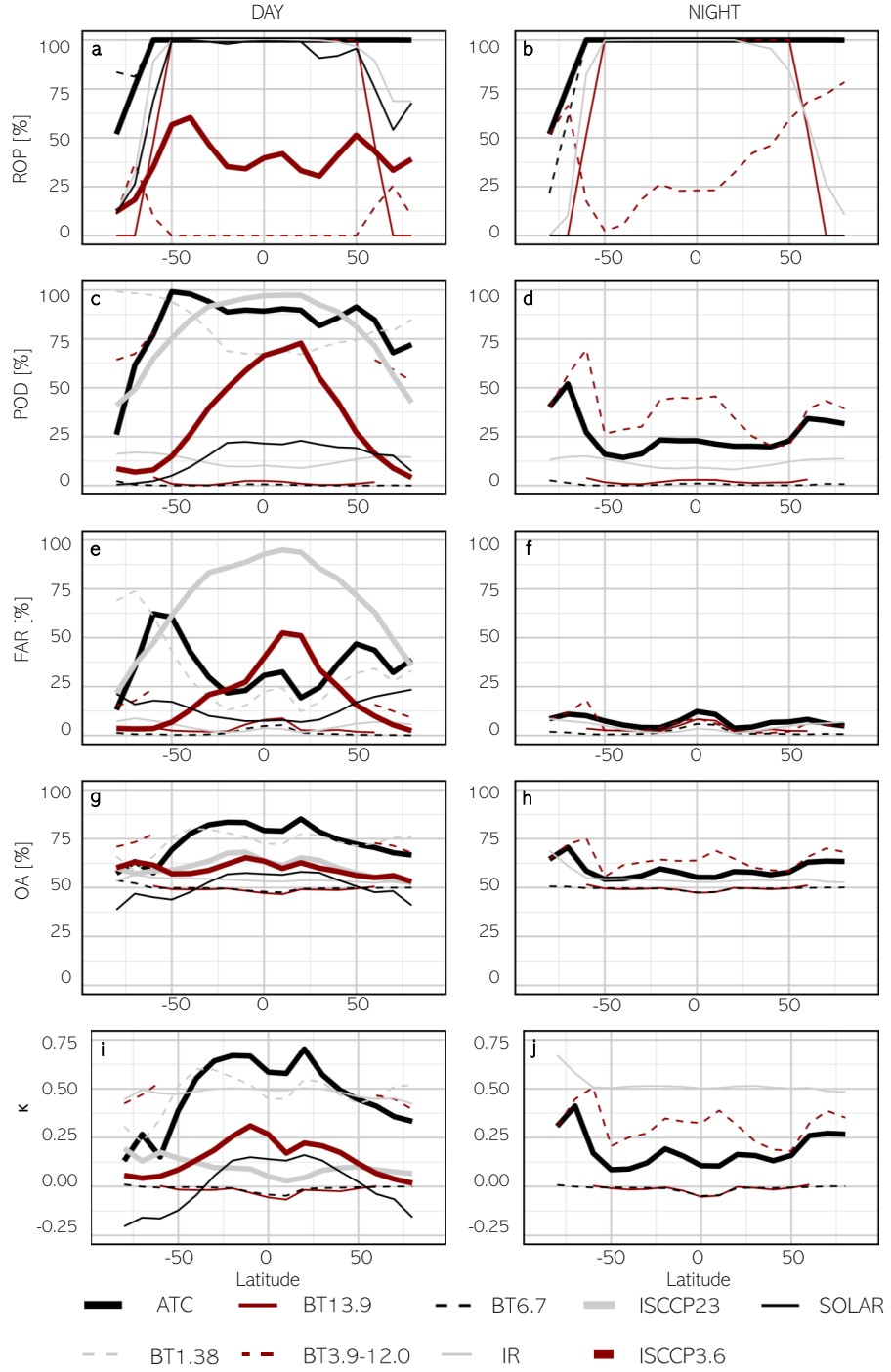

**Figure 5.** Cirrus detection accuracy with respect to the latitude (letters (a, . . . , j) used to facilitate reference in the text)

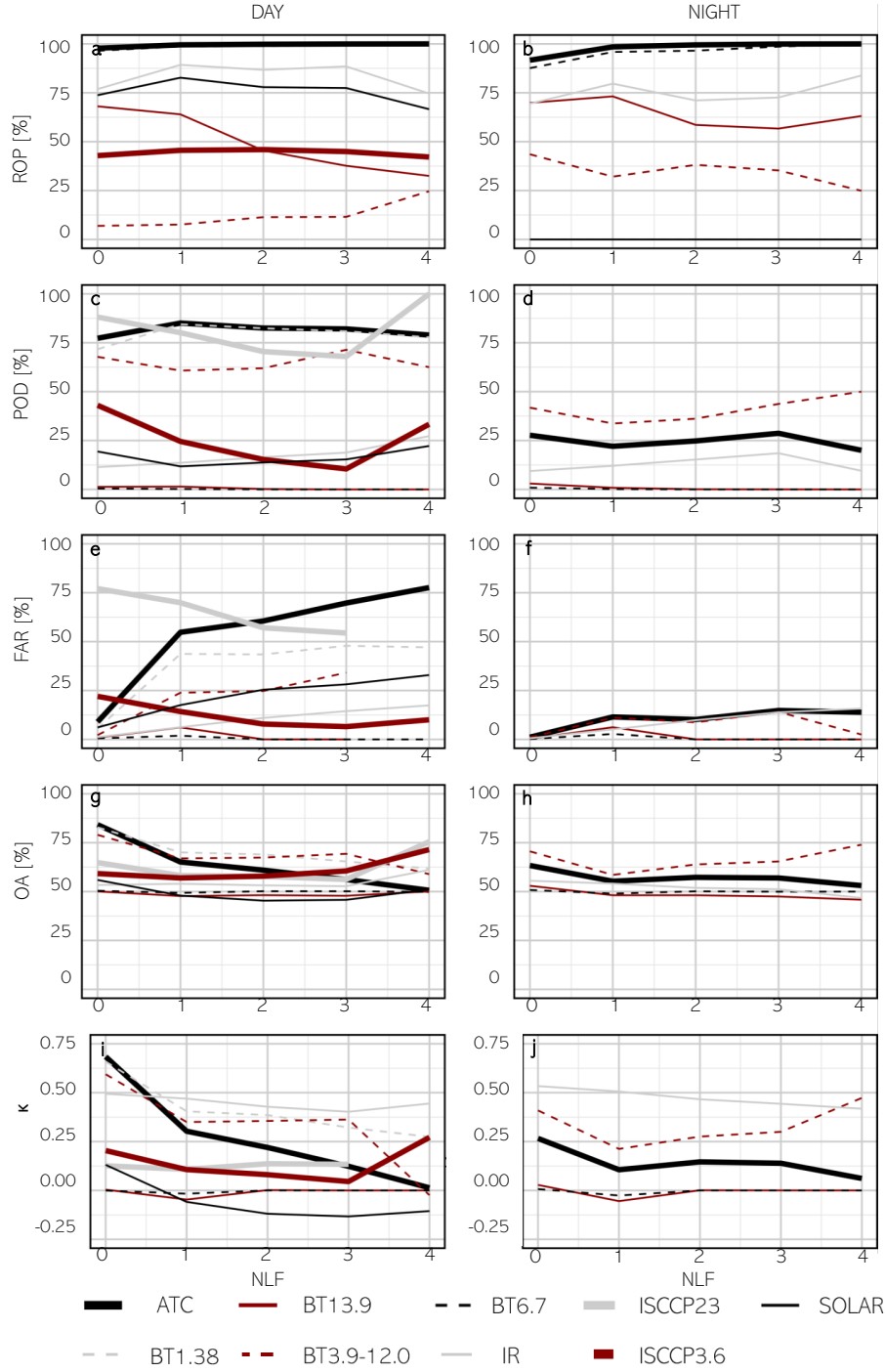

**Figure 6.** Cirrus detection accuracy with respect to the NLF (letters (a, ..., j) used to facilitate reference in the text)

**Table 4.** Goodness-of-fit of cloud detection between MODIS and CALIOP with respect to land classification

| | WATER | | | | | | | | | |
|---|---|---|---|---|---|---|---|---|---|---|
| | **Daytime** | | | | | **Nighttime** | | | | |
| **Test** | **ROP [%]** | **POD** | **FAR** | **OA** | $\kappa$ | **ROP [%]** | **POD** | **FAR** | **OA** | $\kappa$ |
| SOLAR | 88.95 | 11.40 | 13.05 | 49.55 | -0.02 | 0.00 | NA | NA | NA | NA |
| IR | 92.44 | 12.93 | 4.24 | 54.23 | 0.48 | 85.56 | 11.10 | 3.36 | 54.35 | 0.51 |
| BT13.9 | 74.18 | 1.32 | 3.66 | 48.21 | -0.02 | 79.41 | 1.98 | 3.37 | 49.90 | -0.01 |
| BT6.7 | 99.99 | 0.20 | 1.25 | 49.48 | -0.01 | 99.98 | 0.52 | 1.57 | 49.48 | -0.01 |
| BT1.38 | 88.95 | 84.91 | 30.78 | 76.99 | 0.54 | 0.00 | NA | NA | NA | NA |
| BT3.9-12.0 | 5.45 | 67.67 | 16.17 | 74.23 | 0.49 | 14.64 | 51.57 | 8.69 | 70.13 | 0.42 |
| ATC | 100.00 | 90.10 | 40.63 | 74.73 | 0.49 | 99.99 | 18.94 | 6.62 | 56.16 | 0.12 |
| ISCCP23 | 29.32 | 86.27 | 73.48 | 62.45 | 0.14 | 0.00 | NA | NA | NA | NA |
| ISCCP3.6 | 29.32 | 34.69 | 16.22 | 59.89 | 0.19 | 0.00 | NA | NA | NA | NA |
| | LAND | | | | | | | | | |
| | **Daytime** | | | | | **Nighttime** | | | | |
| **Test** | **ROP [%]** | **POD** | **FAR** | **OA** | $\kappa$ | **ROP [%]** | **POD** | **FAR** | **OA** | $\kappa$ |
| SOLAR | 84.11 | 27.39 | 12.70 | 57.53 | 0.15 | 0.00 | NA | NA | NA | NA |
| IR | 93.02 | 11.42 | 4.47 | 52.87 | 0.48 | 80.87 | 9.16 | 2.92 | 53.14 | 0.51 |
| BT13.9 | 77.48 | 1.41 | 3.40 | 49.65 | -0.02 | 86.30 | 2.49 | 3.58 | 49.46 | -0.01 |
| BT6.7 | 100.00 | 0.22 | 1.32 | 49.45 | -0.01 | 100.00 | 0.49 | 1.64 | 49.42 | -0.01 |
| BT1.38 | 88.95 | 84.91 | 30.78 | 76.99 | 0.54 | 0.00 | NA | NA | NA | NA |
| BT3.9-12.0 | 8.09 | 62.80 | 14.99 | 71.91 | 0.45 | 97.78 | 33.85 | 3.61 | 65.15 | 0.30 |
| ATC | 100.00 | 79.62 | 29.87 | 74.88 | 0.50 | 100.00 | 39.34 | 7.80 | 65.77 | 0.32 |
| ISCCP23 | 45.98 | 83.88 | 76.09 | 58.95 | 0.08 | 0.00 | NA | NA | NA | NA |
| ISCCP3.6 | 45.98 | 35.73 | 22.99 | 55.46 | 0.12 | 0.00 | NA | NA | NA | NA |
| | SNOW | | | | | | | | | |
| | **Daytime** | | | | | **Nighttime** | | | | |
| **Test** | **ROP [%]** | **POD** | **FAR** | **OA** | $\kappa$ | **ROP [%]** | **POD** | **FAR** | **OA** | $\kappa$ |
| SOLAR | 10.35 | 6.01 | 20.12 | 41.56 | -0.14 | 0.00 | NA | NA | NA | NA |
| IR | 13.98 | 15.12 | 7.27 | 50.73 | 0.43 | 1.12 | 13.76 | 5.52 | 56.13 | 0.52 |
| BT13.9 | 0.16 | 0.72 | 5.12 | 47.19 | -0.04 | 0.16 | 2.59 | 5.06 | 49.70 | -0.03 |
| BT6.7 | 78.83 | 1.70 | 1.04 | 54.07 | 0.01 | 27.05 | 2.48 | 1.86 | 49.83 | 0.01 |
| BT1.38 | 10.35 | 90.90 | 53.45 | 69.55 | 0.38 | 0.00 | NA | NA | NA | NA |
| BT3.9-12.0 | 13.95 | 61.99 | 15.30 | 69.73 | 0.41 | 47.02 | 39.67 | 7.85 | 65.31 | 0.31 |
| ATC | 88.29 | 27.48 | 10.83 | 59.27 | 0.17 | 55.73 | 33.67 | 7.17 | 62.25 | 0.26 |
| ISCCP23 | 11.34 | 46.54 | 31.07 | 57.85 | 0.16 | 0.00 | NA | NA | NA | NA |
| ISCCP3.6 | 11.34 | 8.00 | 3.64 | 59.62 | 0.05 | 0.00 | NA | NA | NA | NA |

southernmost regions of the Southern Hemisphere are an exception, exhibiting lower values.

Spatial variations in correctly detected cirrus highlight differences between daytime and nighttime POD distribution (Fig. 7c.

430 & Fig. 7d.). During daytime, high values are observed over nearly the entire Earth's surface, with exceptions in Antarctica, Greenland and the Himalayas ($\geq 80\%$ vs $\leq 20\%$ respectively), which are regions covered by snow and ice. However, at night, the highest difference is between land and water ($\geq 50\%$ vs approximately $20\%$). Similar patterns to the POD distribution for day and night can be observed in the OA results (Fig. 7g. - Fig. 7h.). On both sides of the equator, FAR reaches the lowest values, being slightly higher during the day than at night (around $20\%$ and $\leq 5\%$) and increasing with latitude. However,

there is a decrease in FAR observed in regions covered by snow and ice (Fig. 7e. & Fig. 7f.). In regions with the highest rate

of correctly detected and the lowest rate of falsely reported cirrus, the general accuracy of classification (OA) exceeded 80% during daytime and 50% at night. Similar to OA, $\kappa$ was higher during the day. During the day, $\kappa$ values ranged from 0.5 to 1.0 in regions at low latitudes. In contrast, at mid and high latitudes, $\kappa$ values were between 0.0 and 0.5, remaining positive (Fig. 7i.). At night (Fig. 7j.), nearly the entire Earth's surface exhibited $\kappa$ values between 0.0 and 0.5, with negative $\kappa$ values observed near Micronesia.

## 5   Discussion

This study demonstrated that MODIS operational cloud mask product can be used for producing a relatively accurate cirrus mask daytime (73% of overall accuracy, $\kappa = 0.46$), and a questionable quality cirrus mask night-time (60% with $\kappa$ of only 0.2). We suggested the best approach to achieve such a goal, and reported related limitations, specifically for nighttime conditions. During daytime, the two most effective tests were BT1.38 and ATC. With very similar parameters (POD, FAR, OA and $\kappa$) the ATC test demonstrated superiority due to a significantly higher number of available observations. Among the nighttime tests the ATC test proved to be the most suitable for cirrus detection.

Additionally, the ATC test covers nearly all observations in the test (96.7%) and shows relatively low variability of statistical measures across different latitudes. Spatial analysis indicates a very high level of ROP for both: day and night for the entire Earth. Spatial variations observed in correctly detected cirrus highlight differences between daytime and nighttime POD distribution. During the daytime, high values of POD are observed over nearly the entire Earth's surface, with exceptions in the polar regions and Himalayas. However, at night, land regions display higher POD values compared to the surrounding areas.

The ISCCP provides a widely used framework for classifying clouds based on retrieved properties such as optical thickness and cloud-top pressure. In this study, two ISCCP-based classifications were applied using MODIS data: ISCCP3.6, which defines cirrus as clouds with optical thickness < 3.6 and top pressure < 440 hPa, and ISCCP23, which extends the optical thickness threshold to < 23.

ISCCP3.6 showed moderate daytime performance but was limited by a relatively low ROP of 37.97%. ISCCP23 achieved a high POD of 84.16%, but with a correspondingly high FAR of 72.00%, and a slightly better OA of 61.26%. Both tests are only applicable to daytime data and share the same ROP.

These ISCCP-based classifications are included here as a reference framework, not as detection algorithms. Furthermore, the ISCCP classifications used in this study are based on MODIS retrievals, and therefore differ methodologically from early ISCCP climatologies, which relied on AVHRR-based satellite observations.

Considering all mentioned above, the ATC test is proved to be the best among the available methods for detecting cirrus clouds. However, it is evident that its utility during nighttime is limited compared to daytime. A notable factor contributing to this is the sensitivity of CALIOP. Lidar is known to have significantly greater sensitivity at night, which explains its ability to detect nearly twice as many cirrus clouds globally at night compared to daytime. This diurnal pattern in CALIOP data, while highlighting the sensor's advantages in nighttime detection, should not be misinterpreted as a definitive indicator of diurnal differences in cirrus cloud occurrence. Instead, it reflects the increased detection capabilities of CALIOP at night.

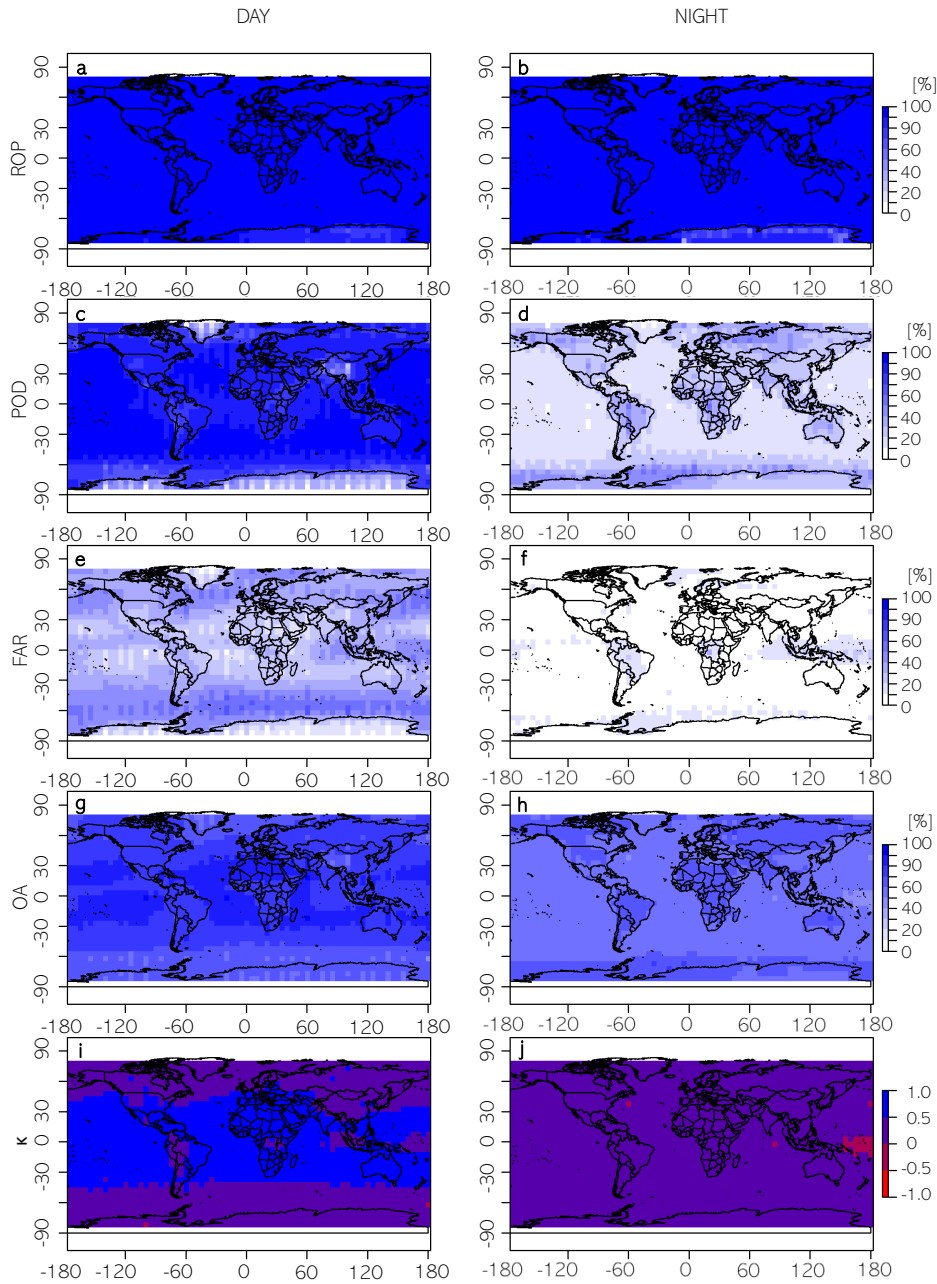

**Figure 7.** Spatial distribution of the accuracy detection of cirrus using ATC test (letters (a, ..., j) used to facilitate reference in the text)

Additionally, MODIS faces further limitations at night due to the unavailability of the 1.38 $\mu$m band (a 'cirrus band', introduced
specifically to detect high, ice clouds; (Gao and Kaufman, 1995), which is highly effective for detecting cirrus clouds during

the day. As shown in the statistical analysis, alternative tests exhibit significantly lower performance compared to the 1.38 $\mu m$
band, emphasizing its critical role in daytime cirrus cloud detection. This limitation further impacts the low effectiveness of
MODIS-based cirrus detection during nighttime observations.

Consequently, we have determined that the ATC test may be suitable for creating a high-level cloud mask and conducting
a long-term climatological analysis of cirrus cloud coverage. This approach simultaneously allows us to address the second
research gap mentioned in this paragraph, which concerns our lack of knowledge regarding the long-term variability of high-
level cloud coverage. Obtained from the CALIOP data, the cirrus mask mentioned in Section 3 allows us to investigate the
distribution of cirrus clouds (Fig. 2.) in 2015. Based on the CALIOP dataset, cirrus cloud coverage reached 18.7% in 2015,
daytime coverage of high-level clouds in 2015 was recorded at 13.2%, whereas nighttime coverage was higher, measured at
480 23.3%. The day-night differences result from CALIOP's higher nighttime sensitivity, reduced lidar signal noise, and increased
nocturnal convective activity leading to more cirrus formation. Additionally, annual variations in cloud amount (over 10 per-
centage points) may occur due to CALIPSO's sampling frequency, as noted by Kotarba and Nguyen Huu (2022).

Similarly, a cirrus mask was generated based on the MODIS data using the ATC test. Derived from this data, cirrus cloud
coverage (Fig. 8a.) showed daytime coverage of high-level clouds at 41.0%, while nighttime coverage was lower, measured at
485 10.9% (Fig. 8b.).

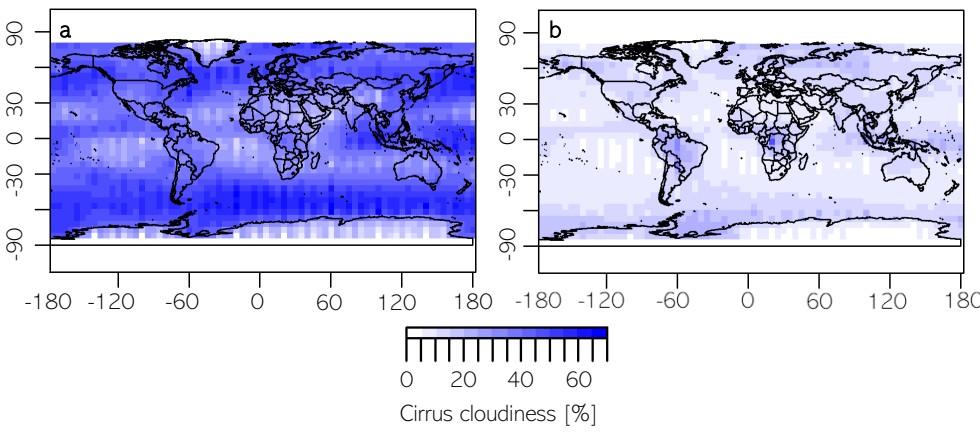

**Figure 8.** MODIS-based cirrus cloud coverage in 2015- daytime (a) and nighttime (b) derived with ATC approach of this study

We also compared cirrus cloud coverage in 2015 obtained from CALIOP and MODIS data (Fig. 9.). Each point in Figure 9
represents mean annual cirrus cloud amount within 5-deg grid box, calculated from all available observations within that grid
cell. The mean difference between cirrus coverage derived from CALIOP and MODIS was -27.71 p.p. for daytime observations
(Fig. 9a.), with MODIS generally indicated higher cloud cover compared to CALIOP. On the contrary, the mean difference
between cirrus coverage derived from CALIOP and MODIS was -12.31 p.p. for the nighttime observations (Fig. 9b.). A linear

regression was performed to evaluate how well MODIS-derived cirrus coverage corresponds to CALIOP values. While the relationship between MODIS and CALIOP is statistically significant (p < 0.001), the $R^2$ value of 0.165 indicates that MODIS captures only 16.5% of the variability. In the nighttime dataset, the $R^2$ improves to 0.422, meaning MODIS cloud coverage

aligns better with CALIOP at night. Additionally, we highlighted in blue the grid cells where the agreement between MODIS and CALIOP cloud classification was moderate or higher, i.e., Cohen's $\kappa \geq 0.5$. During the day, high-$\kappa$ points tend to cluster in regions with low to moderate cirrus amounts. At night, the distribution of high-$\kappa$ points is more dispersed, indicating that even without the 1.38 $\mu$m band, MODIS can achieve strong agreement with CALIOP in selected regions. The divergence between pixel-level classification metrics and the aggregated cirrus cloud cover comparisons arises from differences in MODIS detec-

tion performance between daytime and nighttime conditions. During daytime, MODIS exhibits higher pixel-level agreement with CALIOP, as indicated by elevated $\kappa$ and POD values. However, this is accompanied by a substantial false alarm rate, largely attributable to inherited detections from the 1.38 $\mu$m cirrus test within the MODIS ATC. This spectral test, while highly sensitive to high-level clouds, is prone to overestimations during daytime due to factors such as surface reflection and sun-glint. Consequently, MODIS systematically overestimates cirrus cover relative to CALIOP in daytime aggregated statistics. At night,

the absence of the 1.38 $\mu$m channel reduces the occurrence of false alarms, leading to a closer agreement in total cirrus cover between MODIS and CALIOP, despite the weaker agreement observed at the pixel scale.

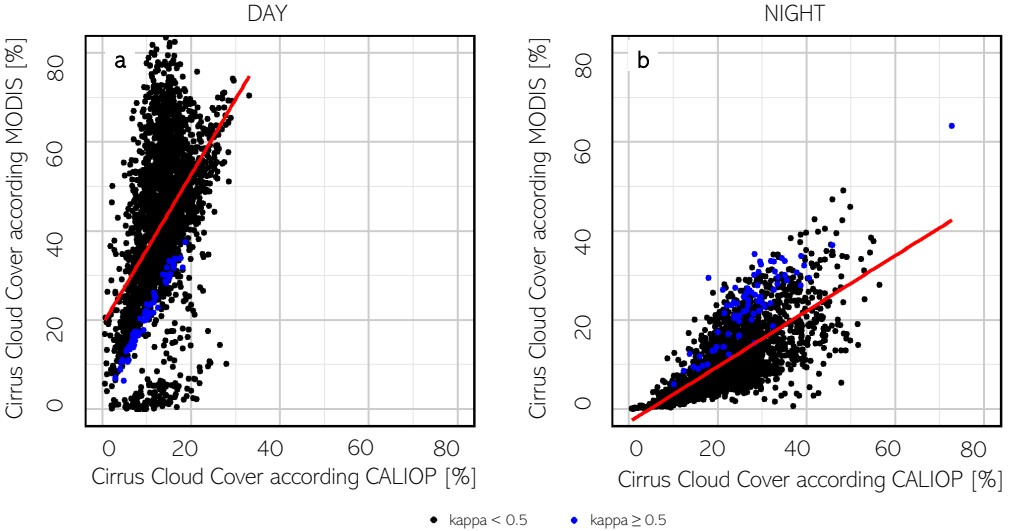

**Figure 9.** CALIOP-MODIS cirrus cloud coverage comparison in 2015- daytime (a) and nighttime (b)

Our goal was to assess the extent to which selected tests of MODIS cloud mask can be used for producing a cirrus mask, acknowledging that MODIS will miss a significant portion of cirrus due to the sensor's lower sensitivity to optically thin clouds.

In order to discuss how the sensitivity possibly impacted our results we examined the MODIS-CALIOP agreement as a function of COT. Information on the cirrus optical thickness was obtained from CALIOP data, and the results are shown in Fig. 10. As observed in Fig. 10., there are no significant changes within the COT range of 0.1 to 1.0, and even up to 10.0. The most noticeable changes occur at COT values close to 10, though these may be influenced by the sample size, as the occurrence of cirrus clouds with a COT near 10 is limited or may represent a misclassification by CALIOP. At such high COT values, the

lidar signal tends to be significantly attenuated, making accurate retrievals increasingly uncertain. This is because as the optical thickness increases, the lidar backscatter signal becomes weaker and may become too weak for precise measurements. Therefore, clouds with such high optical thickness may not be reliably detected by CALIOP, leading to potential misclassifications or missing data (Winker et al., 2024). In some cases, high COT values assigned to cirrus clouds may actually correspond to the cirrus-like top of a strong cumulonimbus cloud, which can be misclassified as cirrus due to the limitations of the CALIOP

classification algorithm under conditions of strong signal attenuation. Notably, differences in parameter values are apparent between a COT of 0 (indicating no cirrus according to CALIOP, at the start of the graph) and 0.1. Upon examining the ATC test results, FAR increases from approximately 30 to 60 during the day, with a similar rise observed at night. The reduced sensitivity of MODIS is reflected in a small but observable increase in POD values as COT increases. Additionally, as thin cirrus clouds become more prevalent, both OA and $\kappa$ values decrease.

As mentioned earlier, CALIPSO can detect cirrus clouds with an optical thickness as low as 0.01, whereas MODIS typically detects them when COT ranges between 0.4 and 0.5. Therefore, we analysed the changes in fit measure as a function of COT within the range of 0 to 1, using a finer step size of 0.01 instead of 0.1 as in previous analyses (Fig. 11.).

During the daytime, most methods show a steady increase in POD as COT rises, while at night, POD also improves significantly with increasing COT, with ATC outperforming other tests. When solar radiation is present, FAR increases with higher

COT for most methods, indicating more false positives as clouds become optically thicker. At night, FAR remains relatively low but shows a slight upward trend with increasing COT. OA remains stable during both day and night. $\kappa$ improves at night for all methods as COT increases but remains lower than daytime values. For daytime, $\kappa$ is highest for ATC and gradually decreases as COT rises.

Given that MODIS inevitably misses a significant portion of cirrus clouds due to its lower sensitivity, we conducted a detailed

analysis for COT values between 0 and 10 and between 0 and 1 with a finer step. The results reveal that fit measures change noticeably with increasing COT for small COT values (<1), a trend that stabilizes for higher COT values. Although it is certain that the issue of thin cirrus clouds generally lowers the fit measures of MODIS to CALIOP, it cannot be said that this is the sole reason for the imperfect fit, as at higher COT values (>1) it also deviates from the full fit. Despite MODIS's limited ability to detect thin cirrus clouds, we do not dismiss its utility. Notably, the ATC method consistently outperforms other approaches

across all evaluated metrics, making it a reliable choice for cirrus detection.

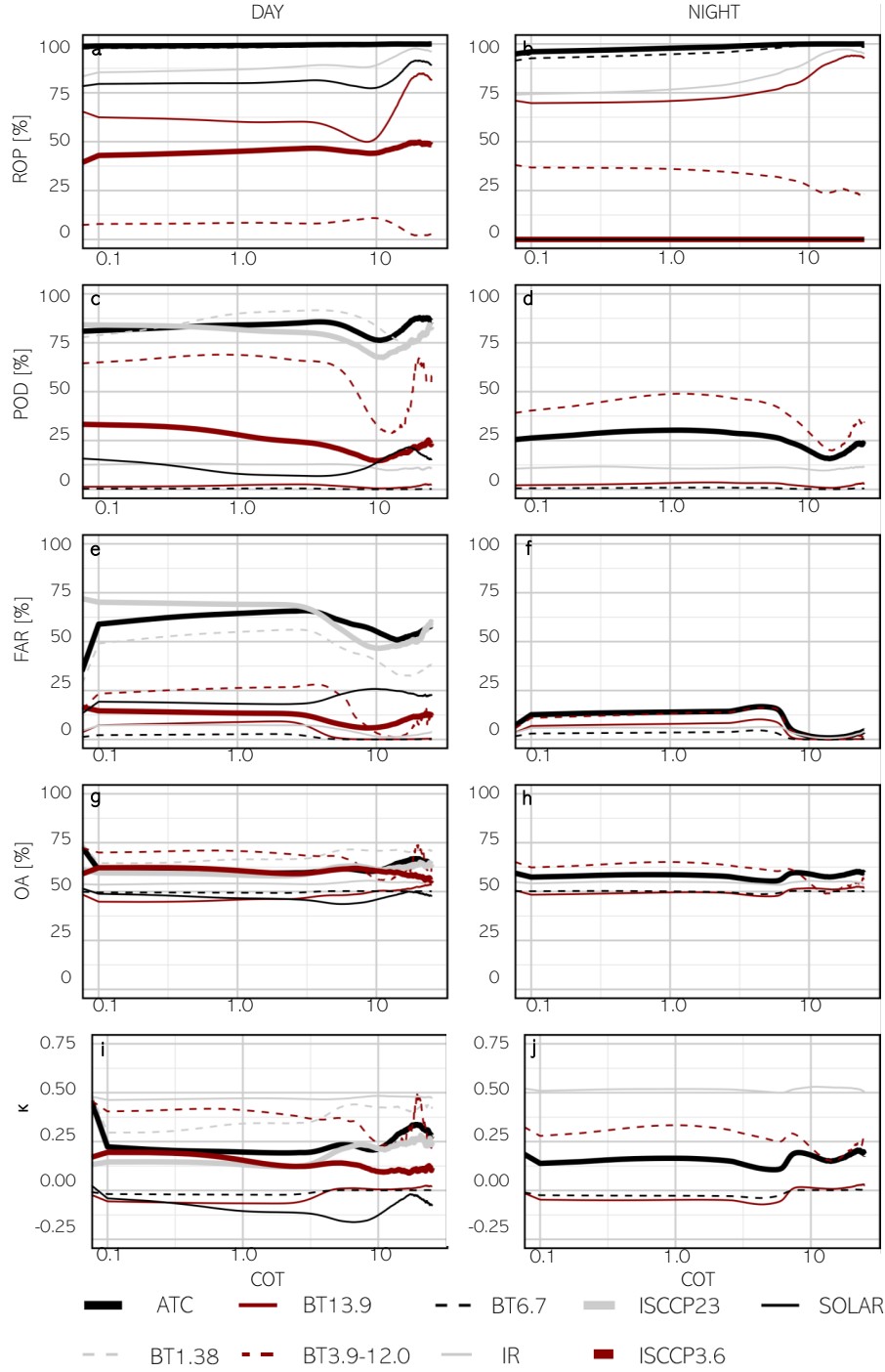

**Figure 10.** Cirrus detection accuracy with respect to the COT (0-25) (letters (a, ..., j) used to facilitate reference in the text)

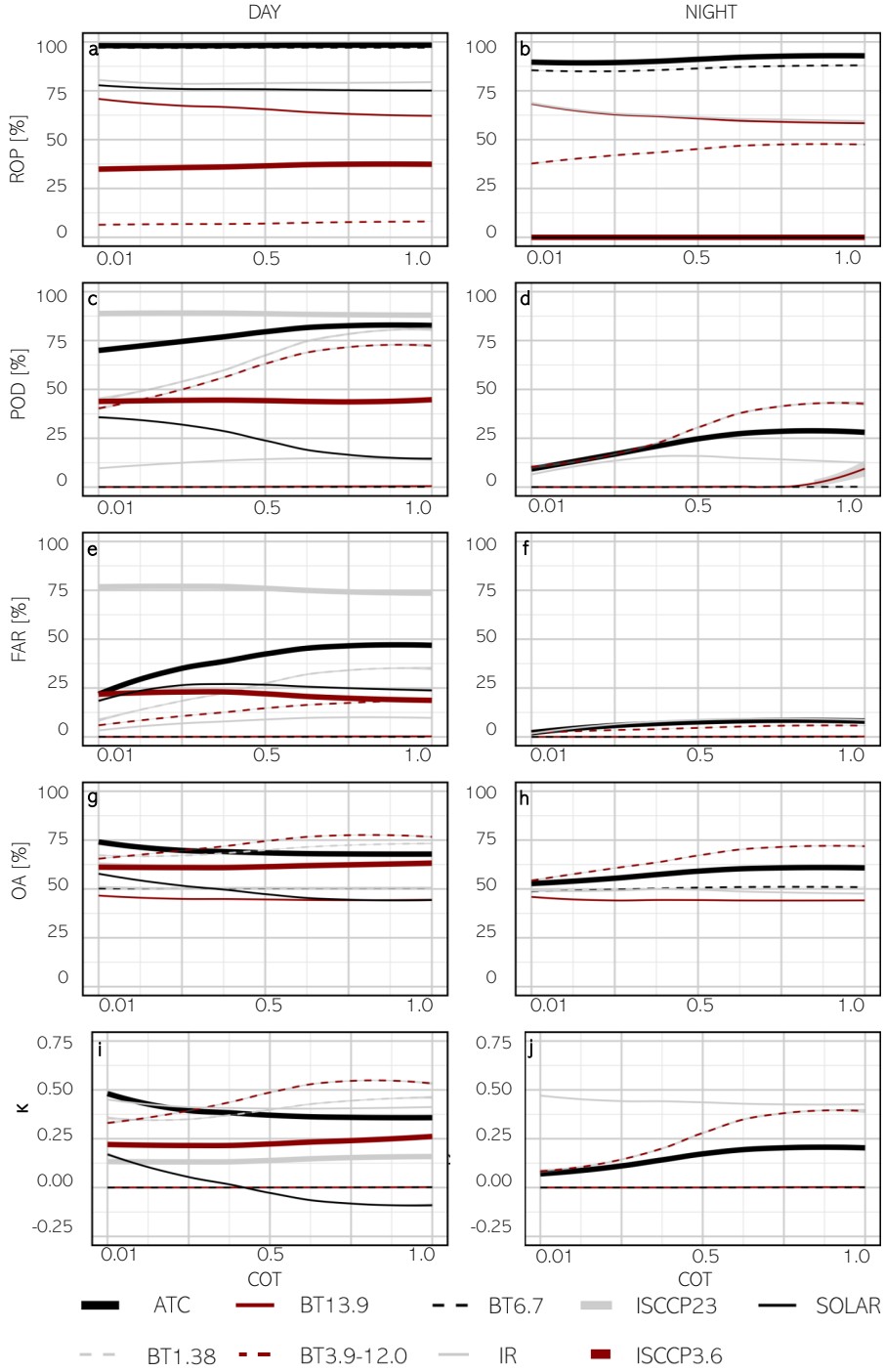

**Figure 11.** Cirrus detection accuracy with respect to the COT (0-1) (letters (a, . . . , j) used to facilitate reference in the text)

## 6 Summary

This study assessed the applicability of the MODIS operational cloud mask product (MYD35) for generating a cirrus cloud mask. To evaluate the accuracy of cirrus detection using the cloud mask tests, we employed a dataset comprising 136 million CALIOP lidar observations from the year 2015 as a reference. The analysis considered: six existing MODIS cloud tests (already reported in the Cloud Mask product), their combination (the ATC test, introduced by this study), and two methods originating from the ISCCP cloud classification scheme.

The key findings was that the ATC test is the most effective for detecting cirrus clouds:

- during daytime, it achieved a moderate reliability, confirmed by an overall accuracy of 72.98%, with a probability of detection (POD) of 80.87%, a false alarm rate (FAR) of 34.86%, and a Cohen's $\kappa$ coefficient of 0.46.

- at nighttime, its showed a low reliability, as proved by an overall accuracy of 59.50%, with a POD of 25.46%, FAR of 6.9%, and low $\kappa$ coefficient of 0.19.

The CALIOP-based cirrus mask revealed a global cirrus cloud coverage of 18.7% in 2015, with higher nighttime coverage (23.3%) compared to daytime (13.2%) due to CALIOP's enhanced nighttime sensitivity. In contrast, the MODIS-based ATC test estimated daytime cirrus coverage at 41.0%, but significantly lower nighttime coverage at 10.9%. Equatorial regions exhibited the highest cirrus frequencies, particularly at night. Although this study is based on one year of data, the large sample size ensures statistical relevance. The ATC test demonstrates relatively high detection capability during daytime and acceptable agreement with CALIOP, but with noted limitations at night and for optically thin cirrus. While MODIS data are often used in cirrus climatologies due to their long-term consistency and global coverage, our findings suggest that cirrus detection within the MODIS cloud mask should be used with caution. The accuracy and reliability observed in this study indicate that the product's applicability to long-term trend analysis may be limited, depending on the specific requirements of the study. In the context of climate studies, the key consideration is not only the absolute accuracy of individual detections, but also the consistency of detection biases over time. Despite its limitations, for the daytime the ATC test shows promise for creating a high-level cloud mask and conducting long-term climatological studies. This study represents a step toward leveraging MODIS data for understanding high-level cloud coverage and its climatic impacts.

*Author contributions.* Żaneta Nguyen Huu: conceptualization, data curation, formal analysis, funding acquisition, investigation, methodology, project administration, software, validation, visualization, writing – original draft preparation & editing

Andrzej Z. Kotarba: conceptualization, data curation, investigation, methodology, writing – review & editing

Agnieszka Wypych: conceptualization, validation, funding acquisition, writing – review & editing

*Competing interests.*   The authors declare that they have no conflict of interest.

# 7   Acknowledgements

The research has been supported by the National Science Center of Poland [grant number 2021/41/N/ST10/02274] and a grant from the Priority Research Area ("Anthropocene") under the Strategic Programme Excellence Initiative at Jagiellonian

University. We gratefully acknowledge Poland's high-performance Infrastructure PLGrid ACK Cyfronet AGH for providing computer facilities and support within computational grant no PLG/2024/016949.

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
