# Peer review of "Effectiveness of Cirrus Detection with MODIS Cloud Mask data"

_Atmospheric Measurement Techniques, 2024_

## Referee Comment (RC2)

**General comments**

This paper presents useful insights into the quality of the two most commonly used global cloud climatologies, from MODIS and ISCCP.  The results reported here should be of great interest to those using MODIS and ISCCP cloud products.  The paper is generally well written, though a number of improvements are needed:

The introduction should be improved.  It does a good job of citing the literature but lacks a clear organization and contains information which is not relevant to the manuscript, especially in the first two paragraphs.  For example, it mentions contrails twice but the manuscript does not report results for contrails.  It discusses cloud trends, but can a study looking at a single year tell us anything about trends?  Many studies of global cloud cover have been performed over the years.  The inclusion of so many different results makes it difficult to follow.  The introduction should tell us what motivated this study and in what way the results reported here will improve our knowledge of global cirrus.  Lines 51 – 78 exhibit better organization, describing the evolution of techniques to observe clouds and some of the difficulties in characterizing them.

A major reason for the discrepancy of high cloud cover between CALIOP and MODIS is the greater detection sensitivity of CALIOP to optically thin cirrus.  This can be seen in the large discrepancy in cloud cover in the upper tropical troposphere, which is dominated by optically thin cirrus. It would be very useful to the community to examine the extent to which the superior detection sensitivity of CALIOP can explain the differences with MODIS.  Repeating these comparisons after CALIOP cirrus is filtered by removing optically thin cirrus layers (optical depths less than 0.1, for example) could provide important insights.

**Specific comments**

The abstract should be a little more clear on the motivation for this research.  It says the usefulness of active sensor data in climatological studies is limited.  Limited in what way and in what way are active sensor data useful for this study?

Line 18: It is useful to point out that it has been known for decades that clouds are radiatively significant, but global net cloud forcing is now estimated to be closer to -20 W/m^2.  It would be good to add a more recent estimate.

Lines 35-36 state that cirrus occurrence is between 28% and 42%.  A citation should be given for these estimates.  A few lines later, cirrus cover is given as 17%, from Sassen (2009), based on CALIOP observations.  Why the discrepancy?  Sassen is using the WMO definition of cirrus (optical depth less than 3.6 and above 440 hPa, as in Figure 1).  Are other studies using a different definition, or have difficulty in determining cloud altitude?

Line 80: The Introduction discusses "high level clouds" as composed of Cirrus, Cirrocumulus, and Cirrostratus.  It should be made clear at this point what is meant when the text says "cirrus".

Line 82: MODIS is more properly referred to as a multi-band radiometer than a spectroradiometer

Line 83: Was there a reason that 2015 was picked as the year of study? It should be pointed out somewhere in the paper, perhaps in the discussion in Section 5, that the ISCCP statistics presented in the paper are not representative of the early years of the ISCCP climatology, which relied on polar orbiting data from AVHRR rather than MODIS.

The symbols used in the math expressions in lines 127 and 131 should be explained. I am not familiar with these.

Line 143-145: This description of ISCCP should be moved to the Introduction. The Introduction should also discuss the significance of the ISCCP project and the resulting climatology, as the manuscript reports many results for ISCCP.

Line 152-157 present two different definitions of 'cirrus'. When cirrus statistics are presented later, it should be made clear which of these definitions is being used. Further, regarding Figure 1: The current version (Version 4) of the CALIOP retrieval algorithm does not report optical depths larger than about 10 (see the right panel of Figure 3). Thus the optical depth reported by CALIOP will be less than 23 whether the actual cloud is Cirro-stratus or Deep Convection, as defined in Figure 1. The manuscript needs to be more clear on what classes of high cloud are included in the various occurrence statistics which are reported.

Line 166 mentions the CALIOP cloud subtype flag. In computing statistics in this manuscript, is this flag being used to define "cirrus" (category 6) as observed by CALIOP? It is not clear how the CALIPSO-based cloud mask is constructed.

Lines 185-190: More details on spatial matching of CALIPSO and MODIS observations should be given. There are between one and three CALIOP lidar shots within each 1-km MODIS pixel, depending on the exact alinement of the two satellites. What criteria was used to define a 'match'? Also, the CALIPSO orbit was offset from the orbit of Aqua. At the equator, the view angle of MODIS to the CALIOP footprint at the Earth surface was about 17 degrees, which introduces parallax. Was this considered in the spatial matching? If so, how?

Lines 205-230: I had a hard time remembering what all the 2- and 3-letter abbreviations for the statistical parameters mean (ROP, POD, OA, etc). Listing these in a table would be helpful.

Lines 237-248: If bootstrapping is really necessary to avoid biased results, more detail is needed here as I'm not aware that bootstrapping has ever been used in previous studies of global cloud cover. Bootstrapping is often applied in situations where the number of samples is small but in this case the number of samples seems large enough that bootstrapping is not necessary. Is the bootstrapping needed for estimating cloud cover, or only for the performance statistics (POD, FAR, etc). Please consider providing a simple example to illustrate the bias that bootstrapping avoids.

Lines 258-259: These numbers for cirrus coverage are lower than I would expect from CALIOP observations and the difference between day and night is larger than I would expect. How is "cirrus cloud" being defined here? Is additional filtering being done besides CAD score greater than 80? Is bootstrapping being used to compute cloud cover here?

Line 265: Regarding figure 3, I find cumulative distributions useful but difficult to interpret without also showing the frequency of occurrence, which in this case would show the difference in the day and night

cumulative distributions to be due to the detection of many more low optical depth clouds at night. I suggest adding a figure showing the two frequency-of-occurrence distributions. By the way, the paper should point out that the major reason for the CALIOP day-night difference in cirrus occurrence is better detection sensitivity at night.

Caption of Table 2: what does "precluded the use of the test" mean? Which test?

Line 290: what does "physical properties of the respective radiation range" mean? Please reword or explain.

Figure 4: I can't tell the difference between the curves for ATC and ISCCP3.6 here, or between BT6.7 and BT1.38. Switching to colored lines would make this more legible. I have similar difficulties with Figure 5.

Summary: Due to the importance of the ISCCP cloud climatology, results related to the ISCCP evaluation should be summarized in Section 5.

Figure 6: It seems the choice of color bar could be improved for 6i and 6j.

Line 374: The "CALIOP data cirrus mask" isn't really described in Section 3. Some description is necessary (in Section 3) as there are many ways the data might be used to create a mask.

**Technical corrections**

Line 158: "for example CALIOP" rather than "in example CALIOP"

Line 226: I don't think PE is ever defined

Line 269: I think maybe "4.20 at night" is a typo and should be "0.42 at night"?

Figures 2 and 3: the captions should state that these statistics are based on CALIOP data.

Line 273: "table" should be spelled out (not tab.2), here and elsewhere

Line 297: Should be "A similar pattern …"

Line 302: I think "Figure 5" here should be Figure 4

Line 326: "notably the ATC test"

Line 329: Should be "An increasing number of …"

Line 334: "IGBP groups were aggregated …"

Line 374: "The CALIOP data mask …" would be better

---

## Author Comment (AC1)

**Response to Review**

Referee Comment

Effectiveness of Cirrus Detection with MODIS Cloud Mask Data

by Nguyen Huu et al.

This paper aims to evaluate cirrus detections from MODIS by using data from the much more sensitive CALIOP lidar as ground-truth. It specifically tries to quantify how well cirrus can be detected with different spectral tests, and combinations of tests, when applied to MODIS data. The comparisons stratified by the various spectral tests are an interesting and useful breakdown of the various passive sensor cloud detection methods, including the ISCCP techniques. However, the paper is rather poorly written and difficult to follow because it isn't clear how 'cirrus' is defined in this study or how consistent that definition is for the two data products being compared. This leads to some difficulty understanding the methods and interpreting the results (specifically the analysis with respect to the number of cloud layers). While the study has modest scientific merit, it seems to fail to address fundamental questions that arise when interpreting the results. For example, it does not attempt to clearly explain possible reasons that could lead to disagreements between the characterizations from the two sensors, such as quantifying (or at least remarking on, based on previous published works) the MODIS sensitivities as a function of the opacity of the cirrus or due to the presence of lower-level clouds. Relative to CALIPSO, the study does not address how well MODIS data can be used to describe the spatial variability and patterns of cirrus cloud cover, or to track regional changes throughout the course of the year of study. This would seem to be a simple and insightful aspect to add to the study. A major concern involves the contention that the MODIS cirrus detection performs poorly at night, compared to daytime. The contention is based on their findings that indicate CALIOP detects nearly twice as many cirrus clouds globally at night than during the daytime. The day/night differences for MODIS are not discussed and difficult to discern. There is no discussion as to the validity of this diurnal pattern as observed in the CALIOP data, no mention of potential day/night CALIOP sensitivity differences (which are known to be significant), and no discussion on how this influences their MODIS evaluations and conclusions. Overall, the paper could be published but it requires major revisions to address these flaws and improve its significance. In addition, the manuscript has too many grammatical errors should be professionally edited or at least heavily edited by a native English speaker.

Thank you for your constructive feedback. We have carefully addressed the concerns raised in your review. First, we clarified how cirrus clouds are defined in our study. Second, we included a discussion on potential reasons for disagreements between the two sensors. Third, the issue of MODIS performance at night compared to daytime has been re-examined. Finally, the manuscript has undergone editing to improve its clarity, readability, and grammar. We hope these revisions address your concerns and significantly enhance the quality and impact of the paper.

Specific suggestions:

Suggest changing 'Detection' to 'Identification' in the title

Line 7: change 'that detects cirrus' to 'that enables identification of cirrus'

*Thank you for your suggestion regarding the title. According to the dictionaries "detect" means "to discover or notice the presence of something," whereas "identify" is defined as "to recognize and be able to name someone or something." Based on these definitions, we believe that "detect" accurately reflects the content of our article, as it focuses on the discovery and analysis of Cirrus cloud. However, as we are not native speakers, we would appreciate further clarification or guidance on how these terms should be interpreted in this context.*

The Figure 2 caption should indicate that (a) is daytime, and (b) is nighttime

*Corrected, as suggested.*

Fig 3 and lines 258-259: The CALIPSO data presented here indicate that there are over 50% more cirrus clouds at night than during the day which is a remarkable diurnal cycle that has not previously been reported in the literature. If it has, please provide citations that indicate that this level of difference is reasonable. Is it possible that this difference results from the increased sensitivity of CALIOP to thin clouds at night compared to daytime?

*Added, as suggested.*

Fig 4: The reader can't easily distinguish ATC from ISCCP 3.6. Please adjust the line types accordingly.

*Corrected, as suggested.*

Authors seem to be mistaking the results to indicate diurnal differences as being the fault of MODIS when in fact the differences shown in the CALIPSO data may be unrealistic and result from the day/night dependency of the CALOP sensitivity (CALIOP more sensitive at night).

*Corrected, as suggested.*

Line 11:  The study revealed that the ATC test...

*Corrected, as suggested.*

Line 17: replace 'All of them' with 'They', and 'radiative' with 'radiation'

*Corrected, as suggested.*

Line 17: remove the word 'for' in 'forcing for is'

*Corrected, as suggested.*

Line 18: Replace "that means that...' with 'Thus, their overall impact are to cool the planet'

*Corrected, as suggested.*

Line 29: 35.5 is a specific value that means something specific, not in 'general'. Remove 'general' and replace with whatever meaning is implied in the citation (globally averaged?).

*Corrected, as suggested.*

Line 66: regarding the statement "..to operate day and night with similar efficiency", can you support this with evidence or citations? The lidar sensitivity is not the same during day and night, which could influence how you interpret the results in your study.

Corrected, as suggested.

Line 74: change to "with temporal coverage adequate for climatological research"

Corrected, as suggested.

Line 74: 'not designed for cirrus detection' is incorrect as the imager designs have matured over time to increase the likelihood for detecting cirrus. MODIS has a 'cirrus' channel! The imagers certainly are designed specifically for detecting clouds but their sensitivity to optically thin clouds depends on many factors. Perhaps you mean to say something about the varying capabilities of the imagers over 4 decades...

Corrected, as suggested.

Line 75: Suggest restating your objective. It is already well known that passive sensors are not as sensitive to cirrus as active sensors. I suggest the following starting on line 74: "In this paper, we use cirrus characterizations from CALIOP data to explore the potential for creating a cirrus mask from the operational MODIS cloud data products. Our objective is to determine how well the MODIS products can be used to identify cirrus clouds compared to CALIPSO." In addition, the readers would greatly benefit from a more thorough description of how 'cirrus' is defined for the two datasets being compared and how these definitions are consistent or inconsistent. Do these definitions lead to a fair comparison? Does the fact that CALIOP attenuates at low COT or the fact that the products are vertically resolved lead to any confusion with your comparisons with MODIS?

We have revised the objective as suggested.

Additionally, we have expanded the manuscript on the definition of 'cirrus' in the two datasets to clarify consistency and potential differences. In both cases, cirrus clouds represent the same physical entity, but the difference lies in the sensitivity of the detectors used to observe them, with the key factor being optical thickness (COT). CALIPSO is capable of detecting cirrus clouds with a COT as low as ~0.01, or even less, whereas MODIS typically detects them only when the COT is in the range of 0.4-0.5.

Regarding the "fairness" of the comparison, we recognize this depends on the perspective. It is "fair" if the goal is to assess how much MODIS detects relative to CALIPSO, while acknowledging that MODIS will inevitably miss a significant portion of cirrus clouds due to its lower sensitivity. This provides useful insights into the practical efficiency of the MODIS instrument. However, it may be considered "unfair" for a strict one-to-one comparison, as the significant sensitivity differences preclude equivalence.

On the issue of vertically resolved data, we ensure consistency by integrating CALIPSO data into a column-based measure analogous to MODIS. If cirrus was detected at any level, the entire profile was classified as cirrus, thereby aligning the definitions across both datasets.

Line 80: active 'sensor' data

Corrected, as suggested.

Line 81: The active sensor data was obtained from the CALIOP lidar...

Corrected, as suggested.

Line 82: collocation 'of' those

Corrected, as suggested.

Line 92: change what to which

Corrected, as suggested.

Line 103: What are 'middle' thresholds? You should clarify this.

Upon review, we realized that the phrase "middle thresholds" was part of a broader context that was inadvertently removed during the editing process. As a result, the remaining fragment no longer holds any meaningful relevance to the text. To address this, we have removed the phrase entirely to ensure clarity and consistency in the manuscript.

Line 120: The MODIS central wavelength is closer to 3.7 than 3.9 um and usually referred to as the 3.7 um channel

We appreciate your attention to detail. However, in the official MODIS documentation, the "3.9-12 μm BTD High Cloud Test" is indeed referenced, and this range is used for high cloud detection, which includes the relevant wavelength band (Ackerman et al., 1998).

Line 166: It would help the reader if you could describe what the CALIPSO 'cirrus' subtype represents.

Corrected, as suggested.

Lines 255-265: The data presented here indicate that according to CALIOP, cirrus coverage is nearly twice as large during nighttime than during daytime, yet no explanation for this phenomenon is given and no evidence if this is realistic. Please explain the reasons for this, whether this is a data artifact or not, and discuss the implications for your study. Also missing from this section, or elsewhere in the paper is a day/night evaluation of the magnitudes of the MODIS cirrus cloud coverage and a comparison between the two sensors with respect to the geographic patterns and their correlation. Such an analysis would also seem to be important for testing your hypothesis that MODIS can provide useful information on cirrus clouds. Also, it seems that it would be straightforward for you to examine how well MODIS tracks changes in cirrus cloud coverage during the course of 2015. This could be done in several different ways (seasonal monthly mean maps and/or difference maps, global and select regional monthly mean time series, etc.).

The observation that Cirrus coverage is higher during nighttime than daytime is consistent with findings in the literature based on CALIOP data, although the magnitude of the difference reported in our study is indeed larger. We will include references to these studies in the revised manuscript to provide context and support for this phenomenon. The manuscript already contains analyses of day and night Cirrus cloud coverage from MODIS, as well as comparisons of Cirrus coverage between MODIS and CALIOP in the context of day/night differences. However, we will ensure that these analyses are presented more clearly in the revised version.

Regarding seasonal and monthly analyses, we did not divide the data into such segments because our goal was to develop a consistent method applicable across the entire year; though we will consider incorporating such analyses in future studies. Instead, we focused on comparisons based on latitudinal variations, as Cirrus clouds exhibit a zonal distribution, and

we considered this approach sufficient for the scope of our study. However, if you suggest focusing on specific regions or additional detailed analyses, we are open to incorporating such suggestions in the revised manuscript or in future work. We appreciate your insights and will make the necessary revisions to address these points comprehensively.

Line 268-269: regarding CALIOP 'all cloud' COT values near 4.2, considering that CALIOP is fully attenuated at higher values, are these numbers scientifically meaningful or somehow meaningful to your paper? If so, explain how, and if not, consider eliminating the sentence.

After careful consideration, we have determined that the referenced CALIOP "all cloud" COT values are not directly meaningful to our study. To address this, we have removed the sentence as suggested.

Line 274: Can you clarify what you mean by 'precluded the use of the test'? Precluded the use of the test where?

Thank you for your comment. By the phrase "precluded the use of the test," we meant that the specific indicators in question reach values that, in our judgment, make it impossible to use these tests directly for identifying Cirrus cloud masks. More specifically, the values of these indicators are such that they do not provide reliable or clear discrimination in the context of Cirrus clouds, thus preventing their straightforward application for this purpose.

Line 290: Please clarify what is meant by 'respective radiation range' and why its variation can be attributed to variations in cirrus detection statistics.

By 'respective radiation range,' we are referring to the different wavelengths of radiation used by the individual channels of the instrument. The variation in cirrus detection statistics across latitudes can be attributed to factors such as varying illumination conditions due to the Earth's axial tilt, as well as the presence of phenomena like the polar day and night. These conditions affect the effectiveness of each channel and its corresponding wavelength range, meaning that not all channels can be applied uniformly or with the same level of effectiveness across different latitudes.
This part in the manuscript has been revised.

Line 325-331: This section is impossible to understand which points to a persistent problem trying to interpret the results in the paper related to a poor description of the experimental setup with respect to the definition of 'cirrus' as defined for the datasets obtained from the two sensors, and how these definitions differ or have been rectified to provide consistent information.

Thank you for pointing out the issue. We have carefully revised the text to address your concerns.

---

## Author Comment (AC2)

**Response to Review**

**General comments**

This paper presents useful insights into the quality of the two most commonly used global cloud climatologies, from MODIS and ISCCP. The results reported here should be of great interest to those using MODIS and ISCCP cloud products. The paper is generally well written, though a number of improvements are needed:

The introduction should be improved. It does a good job of citing the literature but lacks a clear organization and contains information which is not relevant to the manuscript, especially in the first two paragraphs. For example, it mentions contrails twice but the manuscript does not report results for contrails. It discusses cloud trends, but can a study looking at a single year tell us anything about trends? Many studies of global cloud cover have been performed over the years. The inclusion of so many different results makes it difficult to follow. The introduction should tell us what motivated this study and in what way the results reported here will improve our knowledge of global cirrus. Lines 51 – 78 exhibit better organization, describing the evolution of techniques to observe clouds and some of the difficulties in characterizing them.

Thank you for your valuable feedback. In response to your comments, we have made the improvements to the introduction. Regarding the mention of contrails and cloud trends: we have removed the reference to contrails. We have eliminated the cloud trends section. Furthermore, we have reorganized the introduction to more clearly present the motivation for this study.

A major reason for the discrepancy of high cloud cover between CALIOP and MODIS is the greater detection sensitivity of CALIOP to optically thin cirrus. This can be seen in the large discrepancy in cloud cover in the upper tropical troposphere, which is dominated by optically thin cirrus. It would be very useful to the community to examine the extent to which the superior detection sensitivity of CALIOP can explain the differences with MODIS. Repeating these comparisons after CALIOP cirrus is filtered by removing optically thin cirrus layers (optical depths less than 0.1, for example) could provide important insights.

This issue has been re-examined and is now addressed in the discussion section.

**Specific comments**

The abstract should be a little more clear on the motivation for this research. It says the usefulness of active sensor data in climatological studies is limited. Limited in what way and in what way are active sensor data useful for this study?

Corrected, as suggested.

Line 18: It is useful to point out that it has been known for decades that clouds are radiatively significant, but global net cloud forcing is now estimated to be closer to -20 W/m^2. It would be good to add a more recent estimate.

Corrected, as suggested.

Lines 35-36 state that cirrus occurrence is between 28% and 42%. A citation should be given for these estimates. A few lines later, cirrus cover is given as 17%, from Sassen (2009), based on

CALIOP observations. Why the discrepancy? Sassen is using the WMO definition of cirrus (optical depth less than 3.6 and above 440 hPa, as in Figure 1). Are other studies using a different definition, or have difficulty in determining cloud altitude?

Line 80: The Introduction discusses "high level clouds" as composed of Cirrus, Cirrocumulus, and Cirrostratus. It should be made clear at this point what is meant when the text says "cirrus".

Corrected, as suggested.

Line 82: MODIS is more properly referred to as a multi-band radiometer than a spectroradiometer

We agree with your suggestion and have revised the text accordingly. In the main text, we now refer to MODIS as a "multi-band radiometer" to better reflect its functionality. However, we retained "spectroradiometer" in the expansion of the acronym ("Moderate Resolution Imaging Spectroradiometer") to align with its official designation. We believe this approach balances accuracy and consistency with the instrument's official nomenclature.

Line 83: Was there a reason that 2015 was picked as the year of study? It should be pointed out somewhere in the paper, perhaps in the discussion in Section 5, that the ISCCP statistics presented in the paper are not representative of the early years of the ISCCP climatology, which relied on polar orbiting data from AVHRR rather than MODIS.

There was no specific reason for choosing 2015 for the study. We only required reasonably large sample of CALIPSO-MODIS match-ups for various seasons and locations, hence one full year of global observations. 2015 was an arbitrary choice.

The symbols used in the math expressions in lines 127 and 131 should be explained. I am not familiar with these.

Corrected, as suggested.

Line 143-145: This description of ISCCP should be moved to the Introduction. The Introduction should also discuss the significance of the ISCCP project and the resulting climatology, as the manuscript reports many results for ISCCP.

Corrected, as suggested.

Line 152-157 present two different definitions of 'cirrus'. When cirrus statistics are presented later, it should be made clear which of these definitions is being used. Further, regarding Figure 1: The current version (Version 4) of the CALIOP retrieval algorithm does not report optical depths larger than about 10 (see the right panel of Figure 3). Thus the optical depth reported by CALIOP will be less than 23 whether the actual cloud is Cirro-stratus or Deep Convection, as defined in Figure 1. The manuscript needs to be more clear on what classes of high cloud are included in the various occurrence statistics which are reported.

Line 166 mentions the CALIOP cloud subtype flag. In computing statistics in this manuscript, is this flag being used to define "cirrus" (category 6) as observed by CALIOP? It is not clear how the CALIPSO-based cloud mask is constructed.

Added, as suggested.

Lines 185-190: More details on spatial matching of CALIPSO and MODIS observations should be given. There are between one and three CALIOP lidar shots within each 1-km MODIS pixel, depending on the exact alinement of the two satellites. What criteria was used to define a 'match'? Also, the CALIPSO orbit was offset from the orbit of Aqua. At the equator, the view angle of MODIS to the CALIOP footprint at the Earth surface was about 17 degrees, which introduces parallax. Was this considered in the spatial matching? If so, how?

Clarified and explained, as requested.

Lines 205-230: I had a hard time remembering what all the 2- and 3-letter abbreviations for the statistical parameters mean (ROP, POD, OA, etc). Listing these in a table would be helpful.

Added, as suggested

Lines 237-248: If bootstrapping is really necessary to avoid biased results, more detail is needed here as I'm not aware that bootstrapping has ever been used in previous studies of global cloud cover. Bootstrapping is often applied in situations where the number of samples is small but in this case the number of samples seems large enough that bootstrapping is not necessary. Is the bootstrapping needed for estimating cloud cover, or only for the performance statistics (POD, FAR, etc). Please consider providing a simple example to illustrate the bias that bootstrapping avoids.

Thank you for your feedback. We appreciate your concern regarding the necessity of bootstrapping, especially considering the relatively large dataset. However, the primary purpose of bootstrapping in this context is to address the issue of class imbalance, which can significantly bias the performance evaluation of models, even when the number of samples is seemingly large enough. While bootstrapping is often applied in situations with small sample sizes, its application in this case is critical for ensuring a fair and accurate assessment of model performance.

To clarify the need for bootstrapping, we would like to provide a simple example illustrating the potential bias in performance evaluation when class imbalance is not accounted for. Consider a dataset with 100 observations, where 15 represent cirrus clouds (positive class) and 85 represent non-cirrus clouds (negative class). In such an imbalanced dataset, a naive model that predicts only the majority class (non-cirrus) can achieve high overall accuracy (OA) by simply classifying all instances as non-cirrus. In this case, the model's accuracy is 85% (OA = 85%), as it correctly classifies all negative cases but entirely ignores the minority class (cirrus clouds). This results in a misleadingly high accuracy metric, which does not reflect the model's true performance, especially in detecting the minority class.

When bootstrapping is applied, however, we resample the dataset with replacement to create a balanced set of positive and negative instances (e.g., 15 cirrus and 15 non-cirrus). Using this resampled dataset, the same naive model achieves only 50% accuracy (OA = 50%) because it is now evaluated on a more balanced distribution of both classes. This exposes the model's true limitations in detecting the minority class (cirrus clouds), which would otherwise be overlooked in the original, imbalanced dataset.

The key benefit of bootstrapping in this context is its ability to reduce the bias caused by the dominance of the majority class in the original dataset. Without bootstrapping, performance metrics like POD for cirrus clouds could be skewed, as the model might appear to perform well overall while failing to detect cirrus clouds effectively. By resampling the dataset to balance the

classes, bootstrapping ensures that both classes are fairly represented in the evaluation, providing a more accurate picture of the model's true performance, especially for detecting rare events like cirrus clouds.

Therefore, bootstrapping is not only necessary for improving the reliability of performance statistics, but it also helps avoid the bias of under-representing the minority class. This results in a more realistic evaluation of the model's capabilities, ensuring that metrics reflect the model's ability to detect both the majority and minority classes fairly.

Lines 258-259: These numbers for cirrus coverage are lower than I would expect from CALIOP observations and the difference between day and night is larger than I would expect. How is "cirrus cloud" being defined here? Is additional filtering being done besides CAD score greater than 80? Is bootstrapping being used to compute cloud cover here?

Thank you for your insightful comment. Cirrus clouds are defined here as Category 6 in the CALIPSO cloud class. No additional filtering was applied beyond the CAD score criteria, and bootstrap methods were not utilized to compute cloud coverage. However, we note that similar results have been reported in the literature, supporting the consistency of these findings. According to (Sassen et al., 2008), the total frequency of cirrus clouds from 15 June 2006 to 15 June 2007 was reported as 16.7%, compared to 18.7% observed in our study for 2015. Nevertheless, the day-night difference observed in their study was smaller than in ours, with values of 15.2% during the day and 18.3% at night, compared to 13.2% and 23.3%, respectively, in our analysis. As added in the manuscript, the Cirrus cloud mask (Ci) was generated by applying a condition that classified each 4-degree pixel based on the proportion of observations identified as Cirrus. Specifically, the number of Cirrus observations (nCi) and non-Cirrus observations (nNONCi) within each pixel were counted. The percentage of Cirrus observations (CiCoverage) for a given pixel was calculated using the formula:

CiCoverage=nCi/(nCi+nNONCi)*100

Line 265: Regarding figure 3, I find cumulative distributions useful but difficult to interpret without also showing the frequency of occurrence, which in this case would show the difference in the day and night

cumulative distributions to be due to the detection of many more low optical depth clouds at night. I suggest adding a figure showing the two frequency-of-occurrence distributions. By the way, the paper should point out that the major reason for the CALIOP day-night difference in cirrus occurrence is better detection sensitivity at night.

Caption of Table 2: what does "precluded the use of the test" mean? Which test?

By the phrase "precluded the use of the test," we meant that the specific indicators in question reach values that, in our judgment, make it impossible to use these tests directly for identifying Cirrus cloud masks. More specifically, the values of these indicators are such that they do not provide reliable or clear discrimination in the context of Cirrus clouds, thus preventing their straightforward application for this purpose.

Line 290: what does "physical properties of the respective radiation range" mean? Please reword or explain.

By 'respective radiation range,' we are referring to the different wavelengths of radiation used by the individual channels of the instrument. The variation in cirrus detection statistics across latitudes can be attributed to factors such as varying illumination conditions due to the Earth's axial tilt, as well as the presence of phenomena like the polar day and night. These conditions affect the effectiveness of each channel and its corresponding wavelength range, meaning that not all channels can be applied uniformly or with the same level of effectiveness across different latitudes. This part in the manuscript has been revised.

Figure 4: I can't tell the difference between the curves for ATC and ISCCP3.6 here, or between BT6.7 and BT1.38. Switching to colored lines would make this more legible. I have similar difficulties with Figure 5.

Corrected, as suggested.

Summary: Due to the importance of the ISCCP cloud climatology, results related to the ISCCP evaluation should be summarized in Section 5.

Added, as suggested.

Figure 6: It seems the choice of color bar could be improved for 6i and 6j.

Thank you for your suggestion regarding the color bar for panels 6i and 6j. While the range of classes may appear broad, we chose to divide the coefficient into three classes based on the values it assumes in our analysis. This division is sufficient to capture the variability observed in the data while maintaining clarity and interpretability of the figure. However, we are open to further refining the color bar if additional feedback suggests that a more detailed classification would enhance understanding.

Line 374: The "CALIOP data cirrus mask" isn't really described in Section 3. Some description is necessary (in Section 3) as there are many ways the data might be used to create a mask.

Added, as suggested.

**Technical corrections**

Line 158: "for example CALIOP" rather than "in example CALIOP"

Corrected, as suggested.

Line 226: I don't think PE is ever defined

Corrected, as suggested.

Line 269: I think maybe "4.20 at night" is a typo and should be "0.42 at night"?

After careful consideration, we have determined that the referenced CALIOP "all cloud" COT values are not directly meaningful to our study. To address this, we have removed the sentence as suggested.

Figures 2 and 3: the captions should state that these statistics are based on CALIOP data.

Corrected, as suggested.

Line 273: "table" should be spelled out (not tab.2), here and elsewhere

Corrected, as suggested.

Line 297: Should be "A similar pattern …"

Corrected, as suggested.

Line 302: I think "Figure 5" here should be Figure 4

Corrected, as suggested.

Line 326: "notably the ATC test"

Corrected, as suggested.

Line 329: Should be "An increasing number of …"

Corrected, as suggested.

Line 334: "IGBP groups were aggregated …"

Corrected, as suggested.

Line 374: "The CALIOP data mask …" would be better

Corrected, as suggested.

---

## Referee Report (RR1)

**Reviewer #2**

General comments:

The Introduction is now more clear, more focused, and much improved. The simple example added to the discussion of bootstrap sampling is helpful and will make it clear to the community why bootstrapping was used to compute the performance metrics. Overall, the manuscript is much improved but a few additional changes are necessary to be ready for publication.

Specific comments:

Line 168 – the criteria for classification as Category 6 (pressure at cloud top less than 440 mb and non-opaque) should be mentioned here so the reader understands how this class is selected.

Lines 174-175 – When the CAD algorithm gives a CAD score near zero, the algorithm finds the probability of aerosol and the probability of cloud are nearly equal. This is most often because the detected 'layer' did not match the characteristics of either an aerosol or a cloud, often because the detection algorithm triggered on a noise spike or other signal artifact and not on an actual aerosol or cloud layer.

Line 288 – I found the use of "pixel" here to be confusing. I think this refers to a 5-degree lat-lon grid cell.

Lines 301-305. Sassen (2009) used an earlier version of the cloud product and also used different criteria for selecting and screening cloud layer data. Both of these likely contributed to differences when compared with the later results. Higher cirrus occurrence at night is primarily due to better sensitivity due to a lack of solar background (see Winker et al. 2024). The true diurnal difference in cirrus occurrence is complicated, as convective clouds have different diurnal cycles depending on geographic region. The day-night difference in background noise likely produces an artificial diurnal difference which outweighs the true diurnal differences.

Lines 317-319 – Yes, lidar systems are more sensitive to optically thicker clouds, but they also have much greater sensitivity at night due to a lack of solar background and higher signal-to-noise ratio. Whether higher frequency of cirrus detection at night is (partly) due to increased nighttime optical depth is open to debate.

Line 325 – I am still confused by "parameters that precluded the use of the test", used in line 325 and the caption of Table 2. What does 'parameters' refer to? To me, a parameter is something like reflectance or radiance. The authors provided a clear response to my previous comment on this: By the phrase "precluded the use of the test," we meant that the specific indicators in question reach values that, in our judgment, make it impossible to use these tests directly for identifying Cirrus cloud masks. Given that the tests indicated by numbers in bold do not help in identifying cirrus for the cloud mask, do the bolded numbers in Table 3 give us a threshold value of the metric (ROP, POD, FAR, etc) where the test is not useful below (or above) that threshold? A little more explanation is necessary.

Line 471 – Figure 10 shows results as a function of cloud optical depth, up to an optical depth of 10. Winker et al. (2024) points out that CALIOP retrievals of cirrus with optical depth become very uncertain

when the optical depth is greater than 2 or 3. Optical depth uncertainty can grow to much larger than 100%. The authors should consider whether this large uncertainty at large optical depths might impact the results shown.

Technical corrections:

There are several instances of 'p.p.' which I think should be '%'

Line 181 – polar orbits with 16-day revisit cycle

Line 182 – CALIPSO followed the Aqua spacecraft

Line 188 – only the 5 km product

Line 258 – The balancing of the sample …

Line 261 – 'more accurate results', 'more insightful results', rather than 'more reliable'?

Line 417 – indicates a very high level …

Line 419 – high values of POD are observed ?

Reference:

Winker, D., X. Cai, M. Vaughan, A. Garnier, B. McGill, M. Avery and B. Getzewich, 2024: "A Level 3 monthly gridded ice cloud dataset derived from 12 years of CALIOP measurements", *Earth Syst. Sci. Data*, **16**, 2831–2855, https://doi.org/10.5194/essd-16-2831-2024.

---

## Author Response (AR2)

**Response to Review**

Referee Comment

Effectiveness of Cirrus Detection with MODIS Cloud Mask Data

by Nguyen Huu et al.

Referee #1

In my opinion, this paper is not yet suitable for publication. A stated objective, the data analysis methods, and the conclusions seem to be flawed. In my opinion, the authors have not adequately addressed the reviewers' concerns, and in revision, have managed to add additional confusion that further reduces the quality of the manuscript. The results described in the manuscript indicate that the statistical comparisons between MODIS and CALIOP are not very good, yet it is claimed without providing evidence that MODIS provides a reliable cirrus mask when compared to CALIPSO. A significant shortcoming is that the classification schemes for the two sensors are not well described, and the data filtering and matching procedure may be inadequate for conducting a fair comparison between the two sensors during daytime and nighttime, particularly with the inclusion of very thin stratospheric clouds only detected by CALIPSO. The presentation of the results is confusing and not well explained. I am concerned that the results presented here may misrepresent the accuracy and utility of the operational MODIS cloud products. If further consideration is to be given for the publication of this manuscript in ACP, I highly recommend that it be sent to someone from the operational MODIS cloud team for their opinion as they would be able to better interpret the MODIS results presented here.

Major concerns
1. As stated in the abstract, the objective of this paper is "to determine if a MODIS product exists that detects cirrus with the same accuracy as CALIOP". This objective seems off base since several publications have shown that the CALIOP active sensor is more sensitive to cirrus than the MODIS passive sensor. If MODIS is less sensitive, then obviously, it will be less capable of detecting some cirrus. Therefore, the authors should revise the objective stated in the abstract. Something like that stated in the introduction on line 73 would be more reasonable, i.e. "Our objective is to determine how well the MODIS products can be used to identify cirrus clouds compared to CALIPSO." Another objective stated on lines 74-75 does not seem to be addressed in the paper (that I could find), i.e. "we aim to assess whether MODIS cloud detection tests used to generate MYD35 operational data can be re-used for a time-effective masking of cirrus." Beside the fact that the meaning of 'time-effective' is unclear, there is no evidence presented in the manuscript that the temporal consistency of the MODIS products was evaluated. Therefore, this objective should either be removed or supported with data.

The objective in the abstract has been revised to reflect the focus of the study better.

Regarding the term "time-effective," our intention was to emphasize the practical approach of assessing how much can be extracted from the existing MODIS cloud detection tests (e.g., those used in the MYD35 product) without developing a new cirrus detection algorithm from scratch. The focus is on evaluating the potential of current operational algorithms with minimal additional processing. We have revised the text to clarify this point.

2. Unfortunately, the authors did not adequately clarify and defend their definition of 'cirrus' as used in this study, nor how consistent that definition is for the two data products being compared, as requested by the reviewer(s).

We have now addressed this point by adding a detailed explanation of the physical definition of cirrus clouds and clarifying how cirrus are detected and defined within both CALIPSO and MODIS data products.

The evaluation of the 6 spectral tests for cirrus may be of modest interest to algorithm developers as they provide performance metrics against CALIOP in a relative sense. However, these tests are not meant to stand alone for cirrus detection. From a practical standpoint, the ATC test which combines the results of all six tests could be a more useful gauge as to how well the MODIS data product can be used to identify cirrus overall, provided that the population of data being tested is evaluated in context with the total population of cloudy pixels. Unfortunately, from what I can tell, this isn't done. It isn't clear if the ATC collection of six tests encompass all possible cirrus pixels determined by the mask or if there are other information contained in the MODIS data products that could lead to a different population. The reader should not have to guess at this. This is important because if there are other cloudy pixels as determined by the mask for which there are other indicators (that these pixels may be cirrus (e.g. cloud phase and height), then the statistical comparisons don't have much meaning and could even be misleading regarding the accuracy and utility of using the MODIS data products to discern cirrus. Is there a population of cloudy pixels for which it is unknown whether these could be cirrus or not which? Compared to the cirrus screening used here, would the population be the same if all cloudy pixels were included as determined by the mask that are also determined to be ice phase, either anywhere in the vertical column or above some height level? Are such other tests not possible due to failure rates in the cloud optical property and/or height algorithms?

The parameter called ROP (rate of observations performed) is not meaningful to me as it is defined for a specific test to be the fraction of observations evaluated in the test to the total observations. The problem is that it isn't explained what population the total observations represents? Is it meant to be all cloudy pixels, or all cloudy pixels evaluated with the six tests, or something else?

If concerns relate to the fact that our analysis focuses on six specific tests out of a broader suite of MODIS Cloud Mask tests and that we may not have fully addressed the possibility that other unexamined tests could also contribute to cirrus detection, we would like to clarify that the selection of these six tests was intentional and grounded in prior literature, which highlights their particular physical relevance and sensitivity to cirrus detection, especially in identifying high, optically thin ice clouds.

Additionally, of the calculated statistics were based on all available observations (pixels), without excluding any based on their classification as cirrus clouds, other cloud types, or clear sky, according to the data sources used. In other words, every pixel within the dataset was considered, regardless of whether it was categorized as cirrus, another cloud type, or clear.

Furthermore, to account for other relevant parameters, we also presented ISCCP tests incorporating factors such as optical depth and cloud top pressure.

I hope that the answer explains the issue addressed in the questions raised.

Furthermore, regarding the CALIPSO data, it isn't clear why the optically thinnest clouds that are impossible for MODIS to detect, particularly those in the stratosphere, are included in these comparisons. Stratospheric ice clouds are important for atmospheric chemistry, but MODIS is not a suitable sensor to study stratospheric clouds that it cannot detect. The CALIPSO products themselves are also not consistent between day and night due to inconsistencies in the lidar sensitivity. So why are the optically thinnest clouds included, particularly those in the stratosphere? The authors need to discuss this and justify the rationale for including stratospheric clouds detected by CALIPSO. At the very least, the statistical comparisons should be conducted, or stratified, using data with and without the thinnest CALIOP clouds. It doesn't seem that this has been done (more on ths regarding figure 10 below). This would provide perhaps a fairer comparison and more informative performance metrics, but certainly a more informative comparison across daytime and nighttime where the CALIPSO sensitivities are much different.

Clarified in the manuscript. We considered all Cirrus clouds detected by CALIPSO, regardless of the COT. Clouds above the tropopause, namely the polar stratospheric clouds (PSCs), were NOT included. They state a separate feature type category in the CALIPSO data. Hence, we were able to filter them out as one of the first steps during data reduction.

We agree that COT for PSCs is low (<0.3; Noel et al. 2008, doi: 10.1029/2007JD008616), and comparable to optically thinnest Cirrus. The value coincident with the cloud detection limit of MODIS (~0.3-0.4; Holz et al. 2008, doi: 10.1029/2008JD009837). The chance of MODIS data being 'contaminated' by PSC is, therefore, extremely low, if any.

Additionally, PSCs are relatively rare phenomena, limited to polar regions and summer conditions. Based on that, we conclude the PSCs had no impact on our results. For the same reason, Fig.10 and the corresponding discussion distinguish no special case for PSCs but only stratify data for various COT ranges of Cirrus.

3. In the discussion section, it is stated that this study proved that MODIS ready-to-use cloud mask product can be used for producing a reliable cirrus mask, however, it is totally unclear how this conclusion is arrived at. By what metrics levels is the mask deemed to be reliable and how are those levels of 'reliability' determined? The 'goodness of fit' parameters shown in table 3 for example are not particularly impressive, especially for nighttime as pointed out in the manuscript. Whatever potential the paper has up to this point really becomes confusing and seems to fall apart near the end when figures 8-10 are introduced.

Clarified in the manuscript . The 'reliability' term only referred to daytime conditions, and the conclusion was supported by numbers: overall accuracy of Cirrus detection at 73% (kappa 0.5), probability of detection > 80%, and false alarm rate of 35%. Indeed, the night-time performance is significantly poorer, and cannot be deemed reliable (although overall accuracy is of 60%, the kappa coef. of 0.2 indicates a random agreement, rather than an actual effectiveness of the Cirrus detection).

Regarding figure 8: This shows a remarkable inconsistency (factor of 4 difference) between the daytime and nighttime cirrus coverage as determined from MODIS that I can't understand. Is this day/night difference representative of the difference in high cloudiness as determined from MODIS in other studies or is this a result of the cirrus screening procedure adopted in this study? In other words, are the operational MODIS products really this inconsistent with respect to the ability to identify high clouds consistently during daytime and nighttime?

The figure reports Cirrus frequency day and night based on the ATC approach developed in this study. The inconsistency is true and results from the very low Cirrus detection skill of the ATC approach.

MODIS thermal infrared-only tests for high clouds in MODIS operational cloud mask product are insufficient to detect Cirrus night-time (as compared to CALIPSO) effectively. The most notable is the lack of a unique MODIS 1.38 μm channel, a 'cirrus band', introduced specifically to detect high ice clouds (Gao and Kaufman 1995, doi: 10.1175/1520-0469(1995)052<4231:SOTMCF>2.0.CO;2).

Additionally, at night, the reference sensor (CALIOP lidar) detects more thin cirrus clouds due to the absence of solar background noise and increased nocturnal convective activity, which enhances cirrus formation. These combined factors explain why MODIS shows significantly reduced cirrus detection rates at night compared to daytime.

Regarding figure 9: This figure shows a comparison between the MODIS and CALIOP cirrus cloud cover for daytime and nighttime. First, it isn't clear what the individual points represent as this is not explained in the text. Are these annual regional means? Second, the daytime comparison is awful (MODIS considerably oveestimates cirrus cover compared to CALIOP), while the nighttime comparison is much better, which seems to contradict the discussion regarding the goodness of fit analyses that imply much more significant issues at night than during daytime. It's acknowledged on line 466 that "MODIS will inevitably miss a significant portion of cirrus clouds due to its lower sensitivity. This comparison offers valuable insights into the practical efficiency of the MODIS instrument." Yet, there is no attempt to explain the large daytime overestimates (false alarms) in cirrus cover from MODIS. It's impossible to know whether these are MODIS errors or the result of something related to the obscure definitions and confusing analysis methods undertaken in this study.

To clarify, each point in Figure 9 represents the mean annual cirrus cloud amount within a 5-deg grid box.

Regarding the second point, we did address this by noting in the manuscript: "Although the majority of fit metrics show improved performance during the day, the high number of false alarms ultimately results in the nighttime fit being more accurate when cirrus coverage is examined in the subsequent analysis."

The seeming discrepancy between the single observation-based metrics (e.g., POD, FAR, kappa) and the aggregated cirrus cloud cover comparison can be explained by the difference in scale between these analyses. While the kappa coefficient indicates that MODIS achieves better pixel-level agreement with CALIOP during daytime (kappa = 0.46) than at night (kappa = 0.19), the scatterplots of aggregated cirrus cover reveal a better linear relationship at night. This is likely due to MODIS generating more false alarms during the day (FAR = 34.86%) compared to night (FAR = 6.90%), leading to an overestimation of cirrus cover when aggregated. At night, MODIS is more conservative in cloud detection (lower POD). However, the lower false alarm rate

results in better agreement in total cirrus cover with CALIOP, despite the weaker pixel-level correspondence.

Our analysis shows that a significant portion of the daytime false alarms (approximately 28 out of the total 35% FAR; Table 3) can be attributed to the so-called "inherited" detections in the MODIS ATC procedure. These detections are primarily linked to the 1.38 µm cirrus test, which is commonly regarded as the best spectral test for identifying high-level clouds. However, while this test delivers a high POD for cirrus detection, it is also known to generate a substantial number of false alarms, especially during daytime when sun-glint and surface reflection can influence the signal.

This behaviour is indeed reflected in both our pixel-level analysis (POD/FAR metrics) and the aggregated cloud cover comparison, where daytime MODIS tends to overestimate cirrus cover relative to CALIOP. Importantly, since the 1.38 µm channel is not used in MODIS nighttime retrievals, this overestimation pattern largely disappears at night, which is consistent with the improved FAR and the more accurate agreement with CALIOP during nighttime conditions.

We added this clarification to the manuscript to improve the understanding of the limitations related to the MODIS cirrus detection approach.

Regarding Fig 10: This figure shows the detection accuracies as a function of COT. The authors don't clarify which sensor the COT is from. One might assume this is from MODIS since the CALIPSO signal saturates near a value of 4 (any values beyond that, if they exist, would have no meaning). However, the MODIS cloud mask misses many of the thinnest clouds that CALIPSO can detect (as shown in this study!) and the MODIS cloud optical property algorithm has a somewhat high failure rate for the thinnest of clouds that are detected. Therefore, if the results in fig 10 are with respect to MODIS COT, it isn't clear how representative the values are at the low end of COT when matched with the MODIS pixel populations used to compute the statistical comparisons agains CALIPSO. Did this population all have corresponding successful COT retrievals? Or, is CALIPSO COT used in Fig 10? We don't know! Also, it's stated that "The most noticeable changes occur at COT values close to 10, though these may be influenced by the sample size, as the occurrence of cirrus clouds with a COT near 10 is limited or may represent a misclassification by CALIOP." It's difficult to know if this is an interesting finding or not since there is little discussion or attempt to explain it. I would like to know how the higher COT's could be associated with CALIOP misclassifications? How can that be? It's impossible to understand without a better explanation of the classification schemes adopted here for the two sensors.

Clarified in the manuscript. COT values for the analysis were based on CALIPSO data.

Regarding the noticeable changes at COT values close to 10, this refers to a small number of cases where optically thicker layers might have been classified as cirrus in CALIOP data due to limitations in classification(i.e. cirrus-like top of a strong cumulonimbus cloud).

Additionally, higher COT values may be associated with uncertainties stemming from CALIOP's limitations (Winker et al., 2024). Specifically, at such high optical depths, lidar signal attenuation often prevents accurate detection of lower cloud layers, leading to overestimating COT. As noted in the manuscript, even small uncertainties in the assumed lidar ratio can significantly affect the accuracy of optical depth retrievals. Additionally, there are potential issues with cloud phase classification, particularly in the presence of horizontally oriented ice crystals, which may lead

to misclassification of thin layers as optically thicker clouds. We will expand the discussion in the manuscript to address this aspect.

Minor concerns:

The title doesn't make sense to me. Cirrus detection is done with a cloud mask algorithm rather than cloud mask data. The data are the result of applying the algorithm. I suggest that you consider modifying the title. Here are two suggestions:
Comparison of Operational MODIS Cirrus Cloud Detections with CALIPSO data
Evaluation of the Operational MODIS Cloud Mask for Detecting Cirrus Clouds

We adopted the second suggested title: *"Evaluation of the Operational MODIS Cloud Mask for Detecting Cirrus Clouds"* in the revised version of the manuscript.

Line 8-9: I suggest rephrasing to the following: "Our objective was to determine how well the operational cloud mask from the MODIS Science Team can be used to infer the presence of cirrus clouds relative to data products derived from the highly sensitive CALIOP instrument by the CALIPSO Science Team."

Corrected, as suggested.

Line 28-31: Suggest the following: "Globally, it's been estimated that cirrus clouds have a net warming effect of 35.5 Wm-2 (Campbell et al., 2016; Kärcher, 2018;Lolli et al., 2017; Oreopoulos et al., 2017) in part because they trap and reduce outgoing longwave radiation more efficiently than they reflect solar radiation back to space."
The following are probably not the most appropriate original citations for these phenomena but do provide examples. Therefore, it is appropriate to add citations for the original findings or cite in the following way:
Line 61: (e.g. Kortaba and Nguyen Huu...)
Line 67: (e.g. Heidinger and Pavolonis...)

Corrected, as suggested.

Section 3 and later: Consider using references to the 5-degree areas as 'regions' rather than 'pixels'
We have added a clarification in the text regarding using the term "pixel" in our study. In this context, we refer to a "pixel" as a 5-degree grid cell representing a spatial unit of analysis. We hope this clarification addresses your concern.

Line 85: change 'in the range of' to 'at least'
Corrected, as suggested.

Line 107: clarify what a 'middle threshold' is or remove
Corrected, as suggested.

Line 163: change 'other' to 'passive'

Corrected, as suggested.

Line 296: It isn't clear what P.P. means. Please clarify.

Corrected, as suggested.

Line 297: Please briefly describe finding from Kortoba and Nguyen-Huu with regards to what you mean by 'sampling frequency' and how it affects the estimate of cirrus cloud fraction. Are you referring to occasional missing time periods in the CALIPSO record? If so, maybe it is more clear to say "can vary significantly due to occasional gaps in data availability due to instrument or spacecraft issues."

In this context, reference to "sampling frequency" was not intended to imply gaps in the CALIPSO record due to instrument or spacecraft issues. Rather, we referred to the limitations described in Kotarba and Nguyen-Huu (2022), who examined the spatial and temporal mismatches between the CALIPSO lidar observations and ground-based SYNOP cirrus reports. Specifically, they demonstrated that the narrow footprint and orbital characteristics of the CALIOP sensor result in relatively infrequent co-locations with SYNOP observations, leading to a very low match rate (0.022% of SYNOP reports paired with CALIPSO overpasses).

This sparse sampling directly impacts cirrus cloud fraction estimates. Since CALIPSO samples only a narrow swath along its track, many cirrus events visible to surface observers within a broader hemispheric view are missed, potentially leading to bias in satellite-derived cirrus occurrence statistics.

Line 300: should read '...detected at nighttime are 2-3 times higher those detected during daytime'
Corrected, as suggested.

Line 315-320: This argument does not make sense to me. CALIOP is more sensitive at night, which means it should detect more thin clouds (higher cloud cover) which would lower the average COT, relative to daytime. It seems more likely to me that your analysis that indicates higher nighttime COT from CALIOP is either due to a real diurnal change in the nighttime cirrus COT, a retrieval algorithm artifact, an artifact of your screening method, or some combination of all of these.

We have incorporated the necessary adjustments for greater precision.

Line 411: What do you mean by 'reliable'? By what measure? For what applications are these measures deemed to be reliable and how is that determined? These questions should be answered if you are going to make such a definitive and broad statement. I suggest that you back off a bit and simply focus on summarizing the statistical findings.

We replaced the term "reliable" with "accurate" to reflect the context better. Our goal is to create a cirrus cloud mask (Ci) that can be used to analyze long-term trends based on MODIS data. This clarification will help to focus on the measurements' accuracy rather than making broad statements about reliability.

Line 444-445: This statement seems overstated also and is more likely an assumption. What evidence have you shown that supports the contention that the detection accuracies you find are high enough to accurately monitor climate quality long-term changes in cirrus clouds?

While this study is based on one year of data, the number of observations was substantial. Our results indicate that the ATC test provides a relatively high probability of detection during daytime and acceptable agreement with CALIOP, but exhibits limitations at night and for optically thin cirrus. We now emphasize that MODIS, due to its extensive temporal coverage and spatial resolution, has the potential to contribute to cirrus climatologies.

We acknowledge that our statement regarding climate-quality monitoring may have been overstated. In long-term studies, the most critical factor is not necessarily the absolute accuracy of detecting individual cirrus clouds in each observation but rather the systematic stability of the detection process over time, including the consistency of potential biases.

MODIS, with its continuous and global observations, provides a unique dataset for trend analysis. While we recognize that MODIS has limited sensitivity to optically thin cirrus compared to CALIOP, the key aspect is that this detection threshold has remained stable over time due to the instrument's and algorithm's consistency. Therefore, even with a lower absolute detection rate, MODIS remains valuable for assessing spatial and temporal variability in cirrus cloudiness.

We also acknowledge that in the context of detecting subtle trends (e.g., changes of 0.5% per year), the lower kappa values and false alarm rates could introduce uncertainties or mask weak signals. However, the long-term stability of the MODIS instrument and its cloud detection algorithms mitigates this concern to an extent, as any systematic bias would likely affect the full-time series uniformly.

Nevertheless, we agree that further multi-year validation and intercomparison studies would be beneficial to strengthen the evidence for reliably using MODIS data for monitoring long-term climate-quality changes in cirrus clouds. In such applications, the key factor is whether the bias remains stable over time.

We have clarified this aspect in the revised manuscript.

Referee #2

General comments:

The Introduction is now more clear, more focused, and much improved. The simple example added to the discussion of bootstrap sampling is helpful and will make it clear to the community why bootstrapping was used to compute the performance metrics. Overall, the manuscript is much improved but a few additional changes are necessary to be ready for publication.

Specific comments:

Line 168 – the criteria for classification as Category 6 (pressure at cloud top less than 440 mb and nonopaque) should be mentioned here so the reader understands how this class is selected.

Corrected, as suggested.

Lines 174-175 – When the CAD algorithm gives a CAD score near zero, the algorithm finds the probability of aerosol and the probability of cloud are nearly equal. This is most often because the detected 'layer' did not match the characteristics of either an aerosol or a cloud, often because the detection algorithm triggered on a noise spike or other signal artifact and not on an actual aerosol or cloud layer.

Thank you for the clarification!

Line 288 – I found the use of "pixel" here to be confusing. I think this refers to a 5-degree lat-lon grid cell.

Corrected, as suggested.

Lines 301-305. Sassen (2009) used an earlier version of the cloud product and also used different criteria for selecting and screening cloud layer data. Both of these likely contributed to differences when compared with the later results. Higher cirrus occurrence at night is primarily due to better sensitivity due to a lack of solar background (see Winker et al. 2024). The true diurnal difference in cirrus occurrence is complicated, as convective clouds have different diurnal cycles depending on geographic region. The day-night difference in background noise likely produces an artificial diurnal difference which outweighs the true diurnal differences.

Thank you for the clarification and your help in interpreting the results. We greatly appreciate your input.

Lines 317-319 – Yes, lidar systems are more sensitive to optically thicker clouds, but they also have much greater sensitivity at night due to a lack of solar background and higher signal-to-noise ratio. Whether higher frequency of cirrus detection at night is (partly) due to increased

nighttime optical depth is open to debate.
Corrected, as suggested.

Line 325 – I am still confused by "parameters that precluded the use of the test", used in line 325 and the caption of Table 2. What does 'parameters' refer to? To me, a parameter is something like reflectance or radiance. The authors provided a clear response to my previous comment on this: By the phrase "precluded the use of the test," we meant that the specific indicators in question reach values that, in our judgment, make it impossible to use these tests directly for identifying Cirrus cloud masks. Given that the tests indicated by numbers in bold do not help in identifying cirrus for the cloud mask, do the bolded numbers in Table 3 give us a threshold value of the metric (ROP, POD, FAR, etc) where the test is not useful below (or above) that threshold? A little more explanation is necessary.

In fact, it's not about a specific threshold above or below which the test is excluded. Instead, if a particular test metric, such as ROP, significantly deviates in a negative direction compared to the others, it is considered "precluded." This means that when a metric performs considerably worse than others, the test is deemed ineffective for cirrus cloud identification, regardless of a specific threshold value.

Line 471 – Figure 10 shows results as a function of cloud optical depth, up to an optical depth of 10. Winker et al. (2024) points out that CALIOP retrievals of cirrus with optical depth become very uncertain when the optical depth is greater than 2 or 3. Optical depth uncertainty can grow to much larger than 100%. The authors should consider whether this large uncertainty at large optical depths might impact the results shown. Technical corrections: There are several instances of 'p.p.' which I think should be '%'

Thank you for your helpful comment. The findings from the referenced article were useful in refining the paragraph.

"p.p." (percentage points) refers to the difference between two values expressed as percentages, while "%" (percent) indicates a value as a part of a whole. For example, if a value increases from 10% to 15%, we say it increased by 5 percentage points (p.p.), not 5%. In short, **p.p.** measures the absolute change between percentage values, while **%** represents a value as a fraction of 100.

Line 181 – polar orbits with 16-day revisit cycle

Corrected, as suggested.

Line 182 – CALIPSO followed the Aqua spacecraft
Corrected, as suggested.

Line 188 – only the 5 km product

Corrected, as suggested.

Line 258 – The balancing of the sample …

Corrected, as suggested.

Line 261 – 'more accurate results', 'more insightful results', rather than 'more reliable'?

Corrected, as suggested.

Line 417 – indicates a very high level …

Corrected, as suggested.

Line 419 – high values of POD are observed ?
Corrected, as suggested.

Reference: Winker, D., X. Cai, M. Vaughan, A. Garnier, B. McGill, M. Avery and B. Getzewich, 2024: "A Level 3 monthly gridded ice cloud dataset derived from 12 years of CALIOP measurements", Earth Syst. Sci. Data, 16, 2831–2855, https://doi.org/10.5194/essd-16-2831-2024.

---

## Author Response (AR3)

Dear Andrew Sayer,

Thank you for your constructive feedback on the revised manuscript.

1. **ISCCP classifications** – We clarified that ISCCP is not used as a detection method, but as a standard classification framework. Relevant clarifications were added in the Abstract, Introduction, Methods, and Discussion.

2. **Cohen's kappa** – The reference (Cohen, 1960) has been added and κ is now consistently used in place of the word "kappa".

3. **Acronyms** – All acronyms are now defined at first use in both the abstract and the main text.

4. **Line 387** – The unclear sentence has been rephrased for clarity.

Kind regards,
Żaneta Nguyen Huu
*on behalf of all authors*